# COGNATE: Acceleration of Sparse Tensor Programs on Emerging Hardware using Transfer Learning

**Chamika Sudusinghe** [1]  **Gerasimos Gerogiannis** [1]  **Damitha Lenadora** [1]
**Charles Block** [1]  **Josep Torrellas** [1]  **Charith Mendis** [1]

## Abstract

Sparse tensor programs are essential in deep learning and graph analytics, driving the need for optimized processing. To meet this demand, specialized hardware accelerators are being developed. Optimizing these programs for accelerators is challenging for two reasons: program performance is highly sensitive to variations in sparse inputs, and early-stage accelerators rely on expensive simulators. Therefore, ML-based cost models used for optimizing such programs on general-purpose hardware are often ineffective for early-stage accelerators, as they require large datasets for proper training. To this end, we introduce COGNATE, a novel framework that leverages inexpensive data samples from general-purpose hardware (e.g., CPUs) to train cost models, followed by few-shot fine-tuning on emerging hardware. COGNATE exploits the homogeneity of input features across hardware platforms while effectively mitigating heterogeneity, enabling cost model training with just 5% of the data samples needed by accelerator-specific models to achieve comparable performance. We conduct extensive experiments to demonstrate that COGNATE outperforms existing techniques, achieving average speedups of 1.47× (up to 5.46×) for SpMM and 1.39× (up to 4.22×) for SDDMM.

## 1. Introduction

Sparse tensor programs have gained increased significance with the recent advancements in sparse deep learning and graph analytics (Beltagy et al., 2020; Ye & Ji, 2021; Child et al., 2019; Dao et al., 2021) workloads. As a result, many hand-crafted performance optimization techniques have been suggested to improve the performance of sparse

kernels (Kjolstad et al., 2017; Ye et al., 2023; Hong et al., 2019; Jiang et al., 2020). However, the varying non-zero distributions in input sparse matrices have made it difficult to develop performance optimizations for sparse tensor programs that consistently work well across diverse inputs.

To overcome this challenge, machine learning (ML)-based program optimization techniques have been introduced to optimize sparse tensor programs on established hardware platforms (e.g., CPUs) (Won et al., 2023; Yang et al., 2023). These techniques adaptively select a program configuration based on the input sparse matrix features. For example, WACO (Won et al., 2023) introduces learned cost models to predict the runtime cost of programs under different sparse matrices and program configurations. Then, search-based techniques are used to automatically find the optimal program configuration using the cost model's output. Overall, these ML-based techniques show superior performance and adaptability across a diverse range of inputs compared to manually crafted performance optimization techniques.

Recently, on the hardware front, new domain-specific machines specifically designed for sparse operations are emerging (Pal et al., 2018; Hegde et al., 2019; Aananthakrishnan et al., 2023; Gerogiannis et al., 2023; Li et al., 2023; Muñoz-Martínez et al., 2023; Jin et al., 2024). These machines, known as hardware accelerators, offer significant speedups over established hardware platforms. Similar to CPUs, sparse accelerators also provide various program configurations (Gerogiannis et al., 2023; Jin et al., 2024; Gerogiannis et al., 2024), which must be configured by software to achieve the hardware's full potential. It is important that this potential is tested during early-stage hardware development (i.e. before the actual chip is available) to inform better hardware design decisions. For example, hardware architects face the risk of overprovisioning hardware resources (e.g., increasing cache size) to address inefficiencies that could potentially be resolved through improved software strategies (e.g., adopting a better tiling strategy). Therefore, it is crucial to automatically select the optimal program configuration during the design space exploration (DSE) phase of accelerator development.

However, finding the best program configuration for a given

[1]University of Illinois Urbana-Champaign, USA. Correspondence to: Chamika Sudusinghe <chamika2@illinois.edu>.

*Proceedings of the 42nd International Conference on Machine Learning*, Vancouver, Canada. PMLR 267, 2025. Copyright 2025 by the author(s).

early-stage hardware accelerator is challenging. This is because software developers have access only to expensive simulators and, therefore, cannot utilize ML-based cost model development and auto-tuning techniques commonly used for established hardware platforms, which rely on *large* supervised datasets. Such datasets often include hundreds of thousands of labeled examples (Won et al., 2023). Unfortunately, the time needed to collect datasets of similar scale for emerging accelerators relying on simulators is many orders of magnitude longer than the real execution of the program on the actual chip. For example, it can take up to **two weeks to collect a single data point** using the simulator of the state-of-the-art SPADE sparse accelerator (Gerogiannis et al., 2023). At the same time, the same program would take less than a second to execute on the real chip once fabricated. Collecting large datasets would require huge clusters running simulations for months or even years. Therefore, in order to bring the same benefits of ML-based optimizations to accelerator platforms at their early stages, we need to rethink learned cost model construction techniques that are data-frugal and highly sample-efficient.

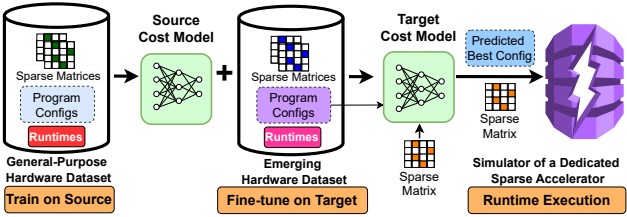

Figure 1: Transfer learning pipeline of COGNATE.

Inspired by the success of transfer learning in other domains (Weiss et al., 2016; Zhuang et al., 2020), researchers have proposed many techniques to reduce data requirements for training cost models (Sasaki et al., 2022; Zheng et al., 2021). These techniques leverage knowledge transferred from cost models learned on one hardware platform (source) to another (target) using the

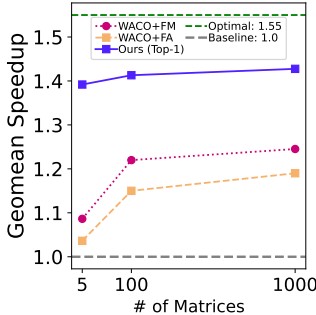

Figure 2: Performance of existing systems for SpMM on SPADE; *WacoNet* with feature augmentation (WACO+FA), and feature mapping (WACO+FM).

ubiquitous pre-train and fine-tune paradigm (Krizhevsky et al., 2012). Such techniques have shown to reduce the data requirement for the target platform. Therefore, using such techniques to transfer learn cost models from general-purpose hardware to emerging accelerators can reduce the data requirements from expensive simulations (Figure 1). However, we notice that most prior works have achieved ef-

fective knowledge transfer only between hardware platforms of *the same type* (e.g. CPU-to-CPU, GPU-to-GPU) (Sasaki et al., 2022; Won et al., 2023; Zheng et al., 2021). Transferring between hardware of different types (e.g. CPU-to-accelerator) poses unique challenges:

**Heterogeneous program configuration spaces.** The program configurations for emerging sparse accelerators, which serve as the input feature space for cost models, can differ significantly from those of general-purpose hardware. For example, emerging sparse accelerators have software-managed buffers instead of hardware-managed caches and specialized, rather than general-purpose, pipelines. This causes a disparity in program configuration spaces for general-purpose hardware and emerging accelerators, making it challenging to naively apply transfer learning. Existing heterogeneous transfer learning techniques (Liang et al., 2019), such as feature augmentation (Daumé III, 2009; Duan et al., 2012), can be a viable approach. However, these techniques often produce feature representations that are too sparse for the cost model to effectively learn, specifically when accommodating a diverse set of program configuration across different hardware platforms. Figure 2 shows the results of applying popular heterogeneous transfer learning techniques – feature augmentation (FA) and feature mapping (FM) – to a learned cost model, WACO (Won et al., 2023). Even when using data samples from 1000 matrices for fine-tuning on the SPADE accelerator, the best configurations found under WACO+FA and WACO+FM are far from optimal. Therefore, we need better techniques to handle the heterogeneity of program configurations across hardware.

**High sample efficiency requirement.** Existing transfer learning solutions for learned cost models operating in homogeneous feature spaces typically require at least 25% of the original dataset used in a non-transfer learning setup to achieve competitive performance on the target hardware platform (Sasaki et al., 2022). The target dataset requirement for these solutions can further increase due to the heterogeneous input feature spaces between general-purpose hardware and emerging accelerators. This makes it infeasible to adopt existing solutions in their current form for accelerators in early design stages. Therefore, we need data-frugal techniques.

**COGNATE.** In this paper, we present COGNATE, a novel framework for developing learned cost models that enable effective knowledge transfer (Figure 1) overcoming these challenges. COGNATE uses WACO's cost model architecture (Won et al., 2023) as the base model (*WacoNet*) but incorporates key changes to make it amenable for transfer learning. This enables the discovery of better program configurations that are closer to the optimal (Figure 2), while requiring significantly less fine-tuning data samples.

COGNATE is centered around two key principles introduced in (Neyshabur et al., 2020): feature reuse and the

capture of low-level statistical information. We observe that, even though the program configurations between general-purpose hardware and accelerators are heterogeneous, there are certain feature spaces that can be mapped due to their similarities. Motivated by this observation, we propose an **approximate mapping of comparable code optimizations**, effectively segregating the feature space generated by program configuration into homogeneous and heterogeneous components. This allows feature reuse across the source and target platforms. The heterogeneous components represent non-mappable hardware specific parameters that can be disparate across different platforms. Such components can introduce challenges during transfer learning due to negative transfer. To separately encode the heterogeneous feature spaces, we introduce a **novel latent space representation of the heterogeneous input feature space** using an auto-encoder. This novel formulation of the feature space allows us to effectively reuse features while minimizing the impact of negative transfer. Further, we observe that certain layers of *WacoNet* do not contribute heavily to the final prediction and this over-parameterization can hinder transferability due to over-fitting. To mitigate this, COGNATE modifies *WacoNet* by reducing the number of layers and extracting features at various depths and scales, effectively allowing the model to capture low-level statistical information.

We evaluate COGNATE on two widely used sparse operations, Sparse Matrix-Matrix Multiplication (**SpMM**) and Sampled Dense-Dense Matrix Multiplication (**SDDMM**). Starting with a CPU as the source hardware platform, we transfer program program configurations to the the state-of-the art SPADE (Gerogiannis et al., 2023) emerging sparse accelerator. SPADE's open ISA and vast set of possible program configurations make it an ideal target platform for our evaluation. Further, to demonstrate the generalizability of COGNATE, we evaluate our techniques for a second target accelerator – an NVIDIA A100 GPU.

Our results show that COGNATE outperforms existing transfer learning techniques by 28.44%, achieving an average speedup of 1.47× (up to 5.46×) for SpMM and 1.39× (up to 4.22×) for SDDMM on SPADE. On the A100 GPU, it attains an average speedup of 1.17× (up to 1.61×) for SpMM and 1.15× (up to 1.49×) for SDDMM.

In summary, this paper makes the following contributions.

- We introduce techniques to segregate and encode the homogeneous (**approximate mapping of comparable code optimizations**) and heterogeneous (**latent representation using an auto-encoder**) components of program configurations across different hardware platforms.
- We introduce COGNATE, a novel data-frugal framework for developing learned cost models that are amenable to few-shot fine-tuning across different hardware platforms, leveraging the above techniques.

- We evaluate and show that COGNATE produces highly accurate transfer learned cost models for emerging sparse accelerators with minimal data collection overhead. Furthermore, we perform extensive experiments and ablation studies to demonstrate its benefits and generalizability.

## 2. Background and Related Work

### 2.1. Sparse Tensor Programs

Sparse tensor programs perform computational tasks that involve tensors where most of the elements are zero. These computations are optimized to efficiently process only the non-zero values. We describe two operations frequently used in these computations below.

**Sparse Matrix-Matrix Multiplication (SpMM)** is the operation of multiplying a sparse matrix $\mathbf{A} \in \mathbb{R}^{M \times K}$ with a dense matrix $\mathbf{B} \in \mathbb{R}^{K \times N}$, resulting in an output matrix $\mathbf{D} \in \mathbb{R}^{M \times N}$. The SpMM operation can be expressed as $D_{i,k} = \sum_j A_{i,j} \cdot B_{j,k}, \quad \text{where} \quad A_{i,j} \neq 0.$

**Sampled Dense-Dense Matrix Multiplication (SDDMM)** is an operation that involves the multiplication of two dense matrices, followed by an elementwise multiplication with a sparse matrix. Given a sparse matrix $\mathbf{A} \in \mathbb{R}^{M \times N}$, a sparse output matrix $\mathbf{D} \in \mathbb{R}^{M \times N}$, and two dense matrices $\mathbf{B} \in \mathbb{R}^{M \times K}$ and $\mathbf{C} \in \mathbb{R}^{K \times N}$, SDDMM operation can be expressed as $D_{i,k} = A_{i,k} \cdot \sum_j (B_{i,j} \cdot C_{j,k}), \quad \text{where} \quad A_{i,k} \neq 0.$

### 2.2. Sparse Tensor Programming Systems

Table 1: Program configuration parameters (Config Params) available across CPU, GPU, and SPADE.

| Config Params | CPU | GPU | SPADE | Type |
|---|---|---|---|---|
| Loop Strip-mining | ✓ | ✓ | | Numerical |
| Loop Reordering | ✓ | ✓ | | Categorical |
| Format Reordering | ✓ | | | Categorical |
| Loop Binding | | ✓ | | Categorical |
| Loop Unrolling | | ✓ | | Categorical |
| Tiling | | | ✓ | Numerical |
| Barrier | | | ✓ | Binary |
| Cache Bypassing | | | ✓ | Binary |
| Matrix Reordering | | | ✓ | Binary |

A sparse tensor programming system supports a range of code optimizations that modify the structure of the code to enhance performance. The effectiveness of these code optimizations depends on assigning specific values to the parameters of the program configuration. By tuning these parameters, we can significantly impact the runtime performance of sparse operations. Table 1 outlines the parameters available for program configurations across different hardware platforms explored in this work (code optimizations are detailed in Appendix B). The execution strategy for sparse tensor programs depends on both the hardware platform and the corresponding programming system used. In this

work, for CPU execution (source platform), we use TACO (Kjolstad et al., 2017), a domain-specific language and a compiler designed for sparse tensor algebra. Considering our target accelerators, SPADE has its own tile-based open instruction set architecture (ISA) to leverage different variations of SpMM and SDDMM operations. For GPU, we employ SparseTIR (Ye et al., 2023), a sparse tensor compilation framework developed as an enhancement to TVM Tensor IR (Chen et al., 2018a).

## 2.3. ML-based cost models

**Learned Cost Models.** Cost models act as fast cost-effective proxies for executing workloads on real hardware. Their primary goal is to accurately estimate the execution time of workloads as they would perform on real hardware. To achieve this, these cost models can be trained on data samples with various program configurations and then be used to predict the program configuration that will deliver the optimal performance. Hence, generally, the training objective of cost models is tied with minimizing $|t^*_{CM} - t^*|$, where $t^*$ is the runtime of the true optimal program configuration and $t^*_{CM}$ is the runtime of the best program configuration suggested by the cost model (***accuracy objective***)(detailed Appendix A). Finding the best configuration suggested by the cost model is usually done using auxiliary intelligent search techniques such as simulated annealing, Monte Carlo tree search, and reinforcement learning. There have been numerous works on learned cost models to predict the runtime of workloads targeting different hardware platforms (Chen et al., 2018b; Adams et al., 2019). These techniques range from simple XGBoost (Chen & Guestrin, 2016) based cost models (Chen et al., 2018b;a) to sophisticated deep neural network based models (Baghdadi et al., 2021; Kaufman et al., 2021; Zhai et al., 2023; Zheng et al., 2020).

**WACO's Cost Model.** WACO (Figure 3(a)) (Won et al., 2023) introduced a learned cost model specifically built for sparse tensor programs, which utilizes sparsity patterns as raw input data. WACO's cost model employs submanifold sparse convolution networks (SCNN) (Graham & Van der Maaten, 2017) to extract features using an *input featurizer*. It leverages a neural network-based *program embedder* to capture the impact of code optimizations on sparse operations by encoding program configurations into embeddings. These program embeddings are merged with the extracted sparsity pattern features produced by the input featurizer. The merged inputs are then processed through a multi-layer perceptron *predictor* to estimate the execution cost.

**Transfer Learning.** Transfer learning is a technique that leverages knowledge gained from a task in a source domain to improve the performance of a related task in a target domain, where data collection can often be challenging (Bozinovski, 2020). There have been many successful examples of transfer learning techniques in a wide range of fields (Weiss et al., 2016). Transfer learning can be catego-

rized into two main types: homogeneous transfer learning (Zhuang et al., 2020), where the input and label spaces are the same, and heterogeneous transfer learning (Day & Khoshgoftaar, 2017), where either one or both can be different. In program optimization, transfer learning has been successfully used to transfer cost models learned from one hardware platform to another, primarily in homogeneous settings, to minimize the target domain data requirements (Zheng et al., 2021; Ryu & Sung, 2021; Sasaki et al., 2022). In this work, we seek to minimize the target domain data requirement during fine-tuning (Shen et al., 2021), by targeting heterogeneous input feature spaces present between general-purpose hardware and emerging sparse accelerators (***data-collection objective***) (detailed in Appendix A).

## 3. Our Methodology: COGNATE

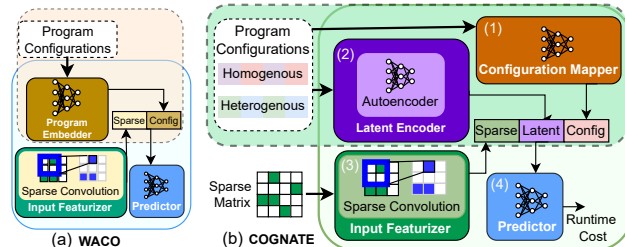

Figure 3: A comparative overview of the enhanced cost model design in COGNATE (b) alongside WACO's cost model design (a), highlighting key differences.

Here, we present COGNATE, a novel framework to design data-frugal learned cost models to accelerate the execution of sparse tensor programs on emerging hardware. The following subsections outline our contributions toward achieving the objectives set forth in Section 2.3; ***maximizing the accuracy*** *while* ***minimizing the data collection overhead***.

### 3.1. Enhancements to Enable Transfer Learning

We build upon WACO considering it as our base model by refining its architecture to better facilitate transfer learning across diverse hardware platforms. These enhancements represent contributions that are orthogonal to WACO's original scope. Our improved cost model design (Figure 3(b)) is structured around four key components: *configuration mapper*, *input featurizer*, *latent encoder*, and *predictor*. The *configuration mapper* (Figure 3(b)(1)) and *latent encoder* (Figure 3(b)(2)) replace the *program embedder* in WACO, while the *input featurizer* (Figure 3(b)(3)) has been modified to more effectively capture low-level information from sparsity patterns. Both *configuration mapper* and *input featurizer* remain consistent across hardware platforms, serving as the components that enable efficient knowledge transfer.

**Configuration Mapper ($\mathcal{FM}$).** The configuration mapper captures homogeneity across hardware platforms by processing program configurations ($c_j$) and their parameters to identify similarities in code optimizations across various

platforms. We designed it to approximately map similar configuration parameters across different hardware platforms (described in Section 3.2 and Section E) to a unified feature space. This is achieved by using explicit mapping functions. The resulting parameters are subsequently passed through a multi-layer perceptron (MLP) to produce the final *configuration vector* $p_j$. In this work, we approximate the code optimizations *loop strip-mining* and *loop reordering* as $p_j = \mathcal{FM}(\phi(\cdot), \pi(\cdot), c_j)$. using the mapping functions $\phi$ and $\pi$, as introduced in Section 3.2.

**Input Featurizer** ($\mathcal{IFE}$). Matrices with identical dimensions and non-zero elements can exhibit vastly different sparsity patterns, making it difficult to extract meaningful features based only on statistical properties. Building on WACO's *input featurizer* (Won et al., 2023), we modify the network architecture (Figure 3) to more effectively capture low-level information from sparsity patterns. Our network consists of 12 SCNN layers (compared to 14 layers in WACO), arranged in 4 blocks, each containing 3 sparse convolution layers. At the end of each block, we apply max pooling to condense spatial information. We increase the number of channels across blocks up to 256, whereas WACO had them fixed at 32. These additional channels enables our design to capture hierarchical features more effectively throughout the network compared to WACO. For a given sparse matrix $M$, our *input featurizer* generates a *sparse feature vector* $s_M$, expressed as $s_M = \mathcal{IFE}(M)$.

**Latent Encoder** ($\mathcal{LE}$). We handle the heterogeneity of program configurations across hardware platforms using per-target autoencoders that compress the heterogeneous components of the configurations into compact latent representations (described in Section 3.3). An autoencoder is trained for each target–sparse primitive pair. During both training and inference, the *latent encoder* $\mathcal{LE}$ processes a configuration ($c_j$), transforming it into a latent representation $z_j = \mathcal{LE}(c_j)$, that encapsulates the unique characteristics of the program configuration.

**Predictor** ($\mathcal{P}$). As the final component of the cost model, the *predictor* (Figure 3(b)(4)) integrates the three feature vectors from the preceding components into a single unified vector, encapsulating all key information about the sparsity pattern and program configuration. This unified vector ($s_M \| p_j \| z_j$) is passed through a multi-layer perceptron (MLP) to eventually output a scalar value representing the predicted execution cost, which can be expressed as $\hat{r}_{M,c_j} = \mathcal{P}(p_j \| s_M \| z_j)$.

### 3.2. Exploiting Homogeneity: Approximate Mapping of Comparable Code Optimizations

Different hardware platforms often use distinct programming systems, leading to variations in how code optimizations are parameterized (Figure 1). Further, an optimization available in one platform may not be directly available on

another, requiring the combination of multiple other code optimizations to replicate the same impact. For example, *loop strip-mining* optimization on CPUs can be closely approximated by collectively applying *barrier* and *tiling* optimizations in SPADE. By mapping the effects of these optimizations using their program configuration parameters, we can expose patterns that facilitate effective knowledge transfer across hardware platforms. In this section, we present our approaches for approximately mapping *loop strip-mining*, *barrier*, and *tiling* optimizations between CPU and SPADE, and *loop reordering* optimization between CPU and GPU.

*Loop strip-mining* is a code optimization that decomposes large software loops into smaller segments to optimize computations for memory utilization and cache performance. In our context, it is applied to loops iterating over the indices $i$, $j$, and $k$ of matrices in SpMM and SDDMM sparse operations (Section 2.1), where parameters $I$, $J$, and $K$ are used to split these loops into outer and inner segments, resulting in a loop nest of six decomposed loops. The resulting loop segments are $\{i_1, i_2, j_1, j_2, k_1, k_2\}$ and their execution order is denoted by $\omega$. In SPADE, we approximate this using *barrier* and *tiling* optimizations. *Tiling* decomposes a matrix into smaller blocks to optimize data reuse in the local memory, while *barrier* controls the execution order of tiles. For example, enabling barrier optimization pauses the tiles scheduled by a control processing element until all previous tiles have been completed (Gerogiannis et al., 2023). Similar to strip-mining parameters, the tiling parameters for $i$, $j$, and $k$ indices of matrices are represented in SPADE as $p_{\text{col}}, p_{\text{row}}, d_{\text{split}}$, respectively, while *barrier* is represented by $b$, where $b = 1$ if barrier is enabled, and $b = 0$ otherwise. Intuitively, tiling divides computations into smaller blocks, while barriers control synchronization during execution. By enabling and disabling barriers for various tiling configurations, we can dictate the order of computation. This resembles loop strip-mining and reordering in CPUs, where optimizing loop execution enhances performance and cache utilization. We can approximately map tiling and barrier parameters to the corresponding strip-mining parameters using the mapping function $\phi : \{p_{\text{col}}, p_{\text{row}}, s_{\text{split}}, b\} \rightarrow \{I, J, K, \omega\}$ as follows:

$$\phi(p_{\text{col}}, p_{\text{row}}, s_{\text{split}}, b) = (I, J, K, \omega)$$
$$I \approx p_{\text{col}}, \ J \approx p_{\text{row}}, \ K \approx s_{\text{split}};$$
$$\omega = \begin{cases} [k_2, j_2, i_2, i_1, j_1, k_1] & \text{if } b = 1 \\ [k_2, i_2, j_2, i_1, j_1, k_1] & \text{if } b = 0 \end{cases}$$

*Loop reordering* is a code optimization that adjusts the execution order ($\omega$) of loops to improve cache efficiency and facilitate parallel processing. It is often applied after loop strip-mining. Here, we examine how it can be approximated for both CPU ($a_1$) and GPU ($a_3$). In CPU, loop strip-mining results in six decomposed loops $\{i_1, i_2, j_1, j_2, k_1, k_2\}$. Similarly, in GPU, loop strip-mining produces six loop segments, but the loop structure differs $\{i_1, i_2, j, k_1, k_2, k_3\}$

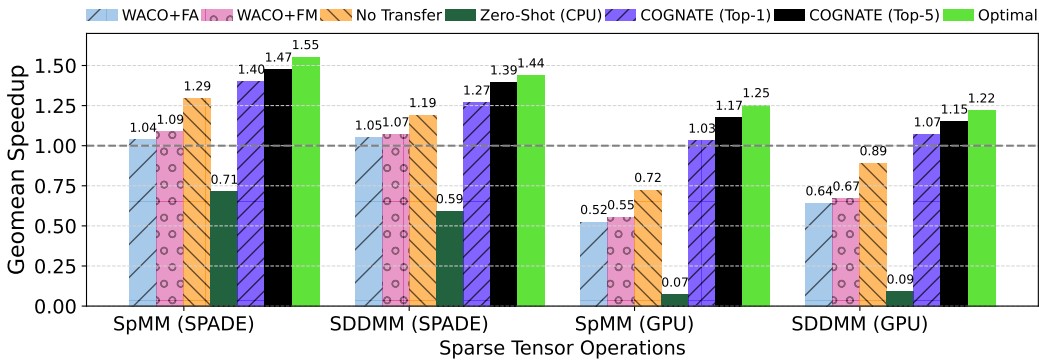

Figure 4: Geomean speedups of COGNATE and other techniques, normalized to the baseline.

due to architectural differences. We approximate them using $\Omega(\cdot)$ function that determines the index of a loop segment and a mapping function $\pi_{a_i} : \{i_1, i_2, \ldots, k_2, \omega_{a_i}\} \rightarrow \{i_1, i_2, \ldots, k_3, \omega'_{a_i}\}$ as follows:

$$\pi_{a_1}(i_1, i_2, j_1, j_2, k_1, k_2, \omega_{a_1}) = (i_1, i_2, j_1, j_2, k_1, k_2, k_3, \omega'_{a_1}) ;$$
$$k_3 = 1, \Omega_{a_1}(k_3) = \Omega_{a_1}(k_2) + 1$$
$$\pi_{a_3}(i_1, i_2, j, k_1, k_2, k_3, \omega_{a_3}) = \left(i_1, i_2, j, j', k_1, k_2, k_3, \omega'_{a_3}\right) ;$$
$$j' = 1, \Omega_{a_3}(j') = \Omega_{a_3}(j) + 1$$

### 3.3. Mitigating Heterogeneity: Latent Encoding of Hardware-specific Code Optimizations

While we can use the strategies described in Section 3.2 to approximate code optimizations with homogeneity, such techniques are not applicable to hardware-specific optimizations. An existing approach for representing hardware-specific optimizations across hardware platforms is to encode them using feature augmentation. However, this results in excessively sparse feature vectors, as code optimizations that are not applicable to a selected hardware platform are zeroed out. Training models on such sparse feature vectors often leads to sub-optimal performance (Figure 4).

To address this limitation, we propose indexing the parameters of the heterogeneous component of the program configurations for each platform $a_i$ using low-dimensional latent representations. Specifically, we train an autoencoder $\mathcal{AE}_{a_i}$ to learn a latent representation $z_j$ for each configuration $c_j \in C_{a_i}$. This is accomplished by determining the value ranges for all parameters of the heterogeneous component in the program configurations, followed by training an autoencoder to learn an unsupervised embedding of this parameterization. Once trained, we use the encoder $\mathcal{LE}_{a_i}$ in $\mathcal{AE}_{a_i}$, which takes each configuration $(c_j)$ as input and transforms it into its corresponding latent representation $z_j$, where $z_j$ is a fixed-size vector. By compressing configurations from different hardware platforms—each with varying parameters and ranges—into fixed-size vectors, we standardize the input for hardware-specific optimizations into the cost model. This compression significantly simplifies the model compared to feature augmentation, as the cost model now processes fewer input features, reducing its computational complexity. With the hardware specific optimizations now represented in a unified latent feature space, it becomes possible to identify and leverage similarities in their impact on performance during fine-tuning. Finally, this approach facilitates the seamless integration of emerging hardware platforms into COGNATE, as we can extend COGNATE to support new target hardware platforms by training new autoencoders and relying on few-shot fine-tuning, eliminating the need to retrain the source model from scratch (detailed in Appendix C). As long as the overall structure of the sparse tensor program remains consistent, COGNATE can quickly adapt by using a small number of new performance samples. In contrast, traditional cost model development approaches would require re-evaluating a large number of configurations (Won et al., 2023).

## 4. Evaluation

### 4.1. Dataset, Training and Evaluation Setup

**Dataset.** Our experiments were conducted using real-world sparse matrices sourced from the SuiteSparse Matrix Collection (Davis & Hu, 2011). This dataset has been widely used in previous work (Pal et al., 2018; Hong et al., 2019; Jiang et al., 2020; Gerogiannis et al., 2023; Won et al., 2023) and covers a broad spectrum of domains, ensuring a realistic and comprehensive evaluation of COGNATE's performance. To collect the training dataset, we performed the sparse operations SpMM and SDDMM on three distinct hardware platforms: an Intel Xeon Gold 6348 **CPU** with 1TB of RAM, an NVIDIA A100 **GPU** paired with an Intel Xeon Platinum 8358, and **SPADE**, a simulated sparse accelerator with 32 processing elements operating at 0.8GHz. To ensure practical feasibility across hardware platforms, the program configuration search space was constrained to 256 configurations for SPADE and approximately 300 configurations for SparseTIR (GPU). We gathered data samples using 1,500 matrices for each hardware platform, with up to 1,000 matrices used for model training under various scenarios and the remainder was set aside for validation. For each matrix, we randomly sampled 100 program configurations per hardware platform to have diverse and representative training datasets across all experiments.

**Program Configuration Search Space for SPADE.** The program configuration search space considered for the SPADE accelerator was derived from a combination of key tunable parameters including tiling, synchronization barriers, cache bypassing, and matrix reordering. As summarized in Table 1, the parameters for barrier insertion, cache bypassing, and matrix reordering are binary (i.e., either enabled or disabled). Tiling is controlled by three numerical parameters: the number of row panels, column panels, and the split factor. These values were chosen to resemble those explored in the original SPADE work (Gerogiannis et al., 2023), as those values were expected to show more significant performance deviations for different sparse matrices. Specifically, we used 4 values for row panels {4, 32, 256, 2048}, 4 values for column panels {1024, 16384, 65536, NUM_MATRIX_COLS} (where NUM_MATRIX_COLS depends on the input matrix) and 2 values for the split factor {32, 256}.

Although COGNATE is designed to perform under limited data availability, we conducted extensive data collection to rigorously evaluate and justify its effectiveness. This included a range of experiments and ablation studies, some of which required performance data samples from up to 1,000 matrices for training. Altogether, this effort demanded approximately 4 million CPU hours. Despite the constrained nature of the search space (256 program configurations), it took nearly three months to complete the dataset curation, even though the simulations were parallelized across multiple machines. Hence, exhaustively evaluating a larger, unconstrained program configuration space would be computationally infeasible. This underscores the need for data-efficient methods like COGNATE, which are designed to operate effectively even under limited data availability.

**Baselines and Implementation.** We executed SpMM and SDDMM on CPU, GPU, and SPADE using the respective programming systems introduced in Section 2.2. We used the default optimizations of these programming systems as our baselines. We implemented COGNATE in PyTorch, utilizing MinkowskiEngine (Choy et al., 2019) to handle sparse convolution. Separate models were developed for SpMM and SDDMM to conduct precise performance predictions. We focused on these two sparse operations because they are the only operations currently supported natively by both the SPADE accelerator (Gerogiannis et al., 2023) and the SparseTIR framework (Ye et al., 2023).

**Cost Model Evaluation.** We evaluated COGNATE's performance on 715 real-world matrices from the SuiteSparse Matrix Collection, ensuring that none of the evaluation data samples overlapped with the training set. For each matrix, we predicted the runtime cost across all program configurations and selected the top-1 and top-5. Then we executed the sparse operations with the selected program configurations on the target platform and identified the one with the short-

est runtime. We then compared our results to the normalized runtime of the baseline executions, *WacoNet* with feature augmentation, and *WacoNet* with feature mapping by calculating the geometric mean (geomean) speedups across matrices to quantify COGNATE's overall effectiveness.

**Pre-training and Fine-tuning Procedure.** The matrices for pre-training were randomly selected from the training set while ensuring a balanced representation of their dimensions and sparsity. To achieve this, we first grouped the matrices into five bins based on the number of rows: ≤8192, ≤32,768, ≤65,536, ≤131,072, and >131,072. From each group, we randomly sampled matrices, ensuring the selected subset collectively spanned a diverse range of sparsity levels. We empirically demonstrate in Section 4.4 (Figure 11) that training the source model with 100 matrices strikes a good balance. We use this setting for our headline result (Figure 4). Once the source model was pre-trained, we performed few-shot fine-tuning on accelerators using data samples from only 5 matrices. This choice was guided by empirical observations, aiming to strike a good balance between transfer learning effectiveness and the cost of data collection. As shown in Section 4.4 (Figure 12), this setting offers the best trade-off between our objectives for accuracy and data collection (detailed in Appendix A). Further, the same set of matrices were used for evaluating the non-transfer learning baselines, enabling consistent and fair comparisons across all experimental settings.

## 4.2. Transferability to SPADE

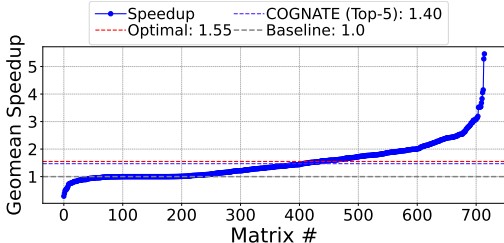

Figure 5: COGNATE per-matrix speedups (SpMM).

Figure 4 illustrates the geomean speedups achieved using multiple techniques: zero-shot inference from the source model (zero-shot), a model trained exclusively on the target hardware using the fine-tuning dataset (no transfer), *WacoNet* with feature augmentation (WACO+FA), *WacoNet* with feature mapping (WACO+FM), and COGNATE's performance for both the top-1 and top-5 (k-best) predicted program configurations. Our results show that COGNATE consistently outperformed other techniques across both sparse operations and hardware platforms. Specifically for SPADE, COGNATE (Top-1) achieved an average speedup of 1.40× for SpMM, reaching 90% of the optimal speedup of 1.55×. When expanding COGNATE (Top-5), it delivered an average speedup of 1.47×, achieving 95% of the

optimal speedup. Note that the optimal speedup was determined by exhaustively evaluating all program configurations within the defined constrained search space for each matrix in the test set, and selecting the fastest configuration per matrix to compute the geometric mean. Similarly, for SD-DMM in SPADE, COGNATE (Top-1) achieved an average speedup of 1.27× and COGNATE (Top-5) achieved an average speedup of 1.39×. This emphasizes COGNATE's ability to consistently find near-optimal program configurations with minimal fine-tuning across multiple sparse operations. The speedup gained for zero-shot inference from the source model was significantly lower than the baseline. In contrast, fine-tuning on a few data samples from SPADE led to significant performance gains showing COGNATE's effectiveness in knowledge transfer.

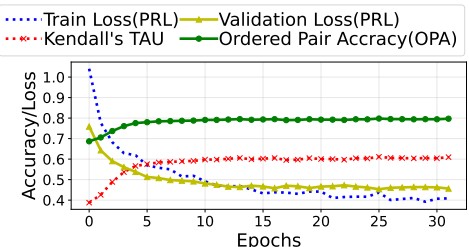

Figure 6: Loss and accuracy during training.

### 4.3. Transferability to GPU

COGNATE is generalizable and is not only applicable to one target hardware platform. To showcase COGNATE's capability, we extended our evaluation to a GPU accelerator (NVIDIA A100) (Figure 4). The speedup trends on GPU aligned with those observed on SPADE, reinforcing the effectiveness of COGNATE. COGNATE (Top-1) delivered an average speedup of 1.03× and COGNATE (Top-5) yielded an average speedup of 1.17× for SpMM, with the optimal achievable speedup being 1.25×. In comparison, cuSPARSE SpMM (Naumov et al., 2010) achieved a lower average speedup of 1.01×. For SDDMM, COGNATE (Top-1) resulted in an average speedup of 1.07×, while COGNATE (Top-1) yielded a 1.15× speedup, with the optimal being 1.22×. Zero-shot inference on the GPU was significantly worse compared to Zero-shot for SPADE, with speedups falling well below the baseline. This discrepancy is likely due to the inherent architectural differences between the CPU and GPU architectures. Further, to assess COGNATE's scalability, we conducted preliminary experiments on an end-to-end GNN workload on GPU. Using the '*transient*' sparse matrix from our test set (178,866 rows/columns (nodes), 961,368 non-zeros) and GraphSAGE model configured with three hidden layers and 256 hidden features, COGNATE achieved a 1.30× speedup for inference and a 1.28× speedup for training over the default SparseTIR (Ye et al., 2023) implementation. These results demonstrate the potential of COGNATE to scale effectively on real-world workloads and deliver consistent performance.

**Comparison with Other Transfer Learning Techniques.** For comparisons, we modified *WacoNet* to support feature augmentation and feature mapping, as it is not inherently optimized for heterogeneous transfer. Despite these modifications, COGNATE consistently outperformed both. For SpMM on SPADE, WACO+FA had an average speedup of 1.04×, while WACO+FM resulted in a slightly higher average speedup of 1.09×. In comparison, COGNATE delivered an average speedup of 1.40×, outperforming its closest alternative (WACO+FM) by 28.44%. The sub-optimal performance of WACO+FA and WACO+FM can be attributed to the increased sparsity in the feature space due to feature augmentation and their limited capacity to effectively mitigate the heterogeneity.

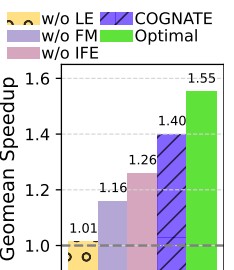 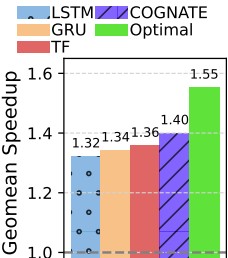

Figure 7: Ablation study of the model components.    Figure 8: Design choices of the predictor.

### 4.4. Additional Experiments for SpMM on SPADE

**Speedup Performance.** Figure 5 shows the speedups achieved by COGNATE (Top-1) across all evaluated matrices. Matrices where the baseline outperformed COGNATE are indicated below the y = 1 dotted line. While the baseline outperformed COGNATE on a few matrices, the overall results demonstrate that COGNATE delivered substantial speedups (as high as 5.46x) for the majority of the dataset.

**Cost Model Accuracy.** Figure 6 shows the accuracy of COGNATE's cost model across training epochs using Pairwise Ranking Loss (PRL), Ordered Pair Accuracy (OPA), and Kendall's Tau (K-Tau). The steady decline in PRL for both training and validation loss indicates that the model effectively learns to rank program configurations without over-fitting. OPA and K-Tau steadily improved to 0.80 and 0.61, indicating effective training.

**Component-Level Contributions.** The effectiveness of our cost model relies on the inclusion of all components, each contributing uniquely to the overall performance. As illustrated in Figure 7, the exclusion of individual components leads to a noticeable decline in speedups. For example, excluding the *input featurizer* ($\mathcal{IFE}$) causes a decline from 1.40x to 1.26x. Similarly, omitting the *configuration mapper* ($\mathcal{FM}$) leads to a further decline to 1.16x, and excluding *latent encoder* ($\mathcal{LE}$) lowers speedup to 1.01x. This emphasizes that each component contributes uniquely to the model's high performance, and all need to act synergistically to maximize the benefits of knowledge transfer.

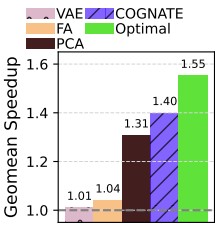

Figure 9: Selection of auto-encoders for COGNATE.

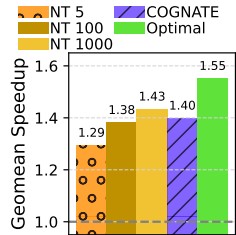

Figure 10: Data overhead w/o transfer learning.

**Selection of MLP Predictor.** As shown in Figure 3, the MLP predictor from WACO's base cost model was retained in our enhanced design. Figure 8 provides a comparative analysis of alternative predictors, including LSTM, GRU, and Transformer (TF). The results demonstrate that our proposed cost model design outperforms the alternatives, with the TF predictor achieving the next best performance with 1.36× speedup. These findings highlight that an MLP predictor is sufficient to deliver robust performance with limited data. In contrast, the suboptimal performance of the TF predictor can be attributed to the limited dataset, as the high simulation costs associated with emerging hardware make it challenging to collect larger datasets for fine-tuning.

**Selection of Auto-Encoders.** Figure 9 shows our investigation into various methods for handling the heterogeneous components of program configurations. We evaluated choices ranging from conventional feature augmentation (FA) to principal component analysis (PCA), auto-encoders, and variational auto-encoders (VAE). Our findings reveal that auto-encoders were the most effective for capturing heterogeneous optimizations in a latent space. This was evident from the lower validation loss observed during the training of the auto-encoders to learn the latent representations.

**Data Collection Overhead w/o Transfer Learning.** Figure 10 shows that without transfer learning, the overhead of data collection becomes significant on emerging hardware due to the high costs of running simulations. For example, models trained exclusively on SPADE would require 20×–200× more target data samples (collected using 100–1000 matrices) to match or surpass the speedups achieved through COGNATE via transfer learning.

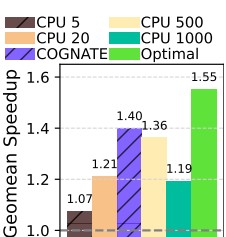

Figure 11: Impact of negative transfer for fine-tuning.

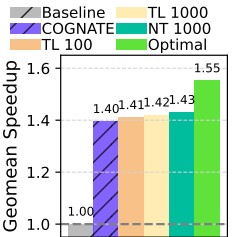

Figure 12: Impact of number of samples for fine-tuning.

**Impact of Negative Transfer.** Figure 11 shows that using a large dataset to train the source model (e.g., data samples

from 1000 matrices) does not necessarily lead to better outcomes. As the size of the training dataset increases, the model becomes overly specialized to the source platform, diminishing its adaptability during fine-tuning. To investigate this effect, we trained source models on datasets of varying sizes (5, 20, 100, 500, and 1,000 matrices) and evaluated their transferability to our target platform (SPADE) using few-shot fine-tuning on just 5 matrices (Figure 11). Our results show that training on the CPU (source) with data samples from 100 matrices and fine-tuning on SPADE (target) with data samples from 5 matrices produces the best results. In contrast, training the source model with data from 1,000 matrices yields sub-optimal performance due to overfitting to source-specific characteristics. This highlights the importance of carefully selecting the size of the source training dataset to avoid over-specialization and minimize the impact of negative transfer.

**Number of Samples in Fine-Tuning.** In Figure 12, we show COGNATE's performance as fine-tuning data samples increase. Despite fine-tuning on data from 1,000 matrices, the maximum speedup saturates at 1.42×. We can achieve a comparable speedup of 1.40× with data from 5 matrices, which shows the diminishing returns associated with larger datasets. Further, the non-transfer learning setup achieved a marginally higher speedup of 1.43× when using data from 1,000 matrices. However, considering the significant data collection overhead, these marginal improvements are not practically justifiable. To further assess sensitivity to the size of the fine-tuning dataset, we conducted additional experiments using 3 and 7 matrices. The resulting speedups were 1.30× and 1.41×, respectively, compared to 1.40× with 5 matrices. While using only 3 matrices led to a noticeable drop in performance, increasing to 7 did not yield a significant gain but required substantially more data samples, incurring several days of additional data collection time. These results suggest that using 5 matrices strikes a practical balance between data collection cost and performance, while demonstrating that COGNATE is relatively robust to small variations in dataset size.

## 5. Conclusion

In this paper, we introduced COGNATE, a novel framework to develop data-frugal learned cost models to optimize sparse tensor programs for emerging hardware platforms. COGNATE leverages a unique technique that capitalizes on the homogeneity of input features across different platforms while effectively mitigating heterogeneity. This enables COGNATE to train cost models using low-cost data samples from widely accessible general-purpose hardware (such as CPUs) and then fine-tune them for emerging hardware platforms with few-shot learning. Our results demonstrate that COGNATE is able to achieve near-optimal accuracy while maintaining significant sample efficiency.

## Acknowledgements

The work is partially supported by the National Science Foundation Graduate Research Fellowship, ACE, one of the seven centers in JUMP 2.0, a Semiconductor Research Corporation (SRC) program sponsored by DARPA, by NSF under the grants CCF-2338739 and CCF-2316233, by DARPA under the grant D24AP00295-00 and by generous gifts from Qualcomm. Any opinion, findings, and conclusions or recommendations expressed in this material are those of the authors(s) and do not necessarily reflect the views of the NSF or DARPA.

## Impact Statement

The goal of this work is to accelerate the execution of sparse tensor programs in the domain of emerging sparse accelerators through the application of machine learning-based techniques. Experiments demonstrate that our approach exhibits potential in benefiting early-stage accelerator development by enabling data-efficient design space exploration. There may be potential societal consequences of our work, none of which we feel must be specifically highlighted here.

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

# A. Problem Formulation

In this work, our aim is to build accurate learned cost models for emerging hardware platforms to enable faster identification of optimal program configurations. A key challenge is the need to maximize the accuracy of the cost model *(accuracy objective)* while using as few expensive (i.e. collected through simulation) data samples as possible *(data collection objective)*. We first formalize the program optimization objective and then tie it with the cost model objectives.

## A.1. Program Optimization Selection

The goal of program optimization in sparse tensor programs is to select the optimal program configuration for a given hardware platform and input sparsity pattern from the total configuration space. Let configuration space $C_a$ be the set $\{c_1, c_2, \ldots, c_{m_a}\}$ of all valid program configurations for a given hardware platform $a$ ($m_a \in \mathbb{Z}^+$). For example, for CPU, a valid configuration from $C_{CPU}$ is a tuple of program configuration parameters for loop strip-mining, loop reordering, and format reordering (Table 1). The optimal program configuration minimizes the execution time of a sparse tensor program. For an input sparse matrix (sparsity pattern) $M$, the optimal program configuration on platform $a$ can be given as, $c^* = \arg\min_{c_i \in C_a} \mathcal{T}_a(M, c_i)$, where $\mathcal{T}_a$ is the execution time function for platform $a$ (ground truth runtime). The execution time for the optimal program configuration is given by $t^* = \mathcal{T}_a(M_l, c^*)$.

## A.2. Cost Model Performance and Data Efficiency Objectives

We approximate the ground truth runtime $\mathcal{T}_a$ using learned cost models. Usually, these cost models are trained with one objective: to achieve high accuracy. However, due to the high cost of simulation in emerging hardware, we also want to minimize the amount of data samples required from these platforms for model training. We formalize these two objectives as follows.

**Data Collection Objective (DCE).** Let $D_a = \{(M_l, c_i), t_i \mid i \in m_a, l \in \mathbb{Z}^+\}$ be the dataset collected from hardware platform $a$, and let $\beta_a$ represent the average cost of collecting a single data sample from the platform. Our objective is to $\min_{D_a} \beta_a \times |D_a|$.

**Accuracy Objective.** Let $CM_a$ (which approximates $\mathcal{T}_a$) be the learned cost model trained on dataset $D_a$. If the best program configuration returned by the cost model ($c^*_{CM_a}$) has an actual execution time $t^*_{CM_a}$, our objective is to $\min |t^*_{CM_a} - t^*|$, where $t^*$ is the execution time for the optimal configuration. For a set of input sparse matrices $\{M_1, M_2, \ldots, M_k\}$, our objective can be extended to minimizing the Absolute Percentage Error (APE) across all matrices:

$$APE = \frac{1}{k} \sum_{l=1}^{k} \frac{|t^*_{CM_a, M_l} - t^*_{M_l}|}{t^*_{M_l}} \times 100$$

where $t^*_{CM_a, M_l}$ denotes the execution times for the predicted best program configuration for the input sparse matrix $M_l$ and $t^*_{M_l}$ denote the optimal program configurations for the same matrix.

## A.3. Evaluations for Cost Model Objectives

To evaluate the cost model objectives, we conducted the following experiments for SpMM on SPADE. For simplicity in the calculations, we set $\beta_{\text{CPU}} = 1$ and $\beta_{\text{SPADE}} = 1000$. However, a CPU execution typically takes milliseconds, while a SPADE execution can extend up to two weeks. We explored 11 distinct models across four different categories, differentiated by the number of data samples they were trained on, while the cost model architecture remained the same. Category I consists of models (NT $d$) trained exclusively on data samples from $d$ matrices executed on SPADE. Category II includes transfer-learned models (TL $d$), which were pre-trained with data samples from 100 matrices on CPU (10,000 data samples) and then fine-tuned on SPADE with data samples from $d$ matrices. Category III consists of models (CPU $d$) pre-trained with varying numbers of data samples from $d$ matrices on CPU and then fine-tuned on data samples from 5 matrices on SPADE. Finally, we did zero-shot inference (Zero-Shot) from a model pre-trained on CPU with data samples from 100 matrices without additional fine-tuning on SPADE.

Models trained exclusively on SPADE data samples (NT d) generally exhibit increasing speedup and decreasing APE as the number of SPADE data samples increases. For example, NT 1000, trained on 100,000 SPADE data samples, achieves the highest speedup of 1.43 and an APE of 7.06. However, the data collection overhead for these models rises significantly with the number of SPADE samples, making the use of them impractical due to the long simulation times. In contrast, the TL

models, which are pre-trained on CPU data and fine-tuned on SPADE samples, demonstrate an excellent balance between speedup, APE, and DCE. TL 5 model, for instance, delivers a competitive speedup of 1.40 and a low APE of 7.28, while maintaining an excellent DCE of 0.51.

| Model | Data Samples | | COGNATE (Top-1) Speedup | APE | $DCE/10^6$ |
|-------|------|-------|---------|-----|-----|
| | CPU | SPADE | | | |
| NT 5 | - | 500 | 1.29 | 15.02 | 0.50 |
| NT 100 | - | 10000 | 1.38 | 9.42 | 10.00 |
| NT 1000 | - | 100000 | 1.43 | 7.06 | 100.00 |
| TL 5 (CPU 100) | 10000 | 500 | 1.40 | 9.58 | 0.51 |
| TL 100 | 10000 | 10000 | 1.41 | 8.74 | 10.01 |
| TL 1000 | 10000 | 100000 | 1.42 | 7.28 | 100.01 |
| CPU 5 | 500 | 500 | 1.07 | 27.80 | 0.50 |
| CPU 20 | 2000 | 500 | 1.21 | 19.35 | 0.50 |
| CPU 500 | 50000 | 500 | 1.36 | 16.34 | 0.55 |
| CPU 1000 | 100000 | 500 | 1.19 | 36.00 | 0.60 |
| Zero-Shot (CPU) | 10000 | - | 0.71 | 46.22 | 0.01 |

Table 2: Comparison of cost model performance with varying data samples from CPU and SPADE.

## A.4. Learning Objective

Our objective is to train a cost model that effectively learns to identify a program configuration that minimizes the runtime of a sparse operation for a given sparsity pattern. To achieve this, we begin by training our cost model to learn the relative rankings of program configurations during execution, enabling it to accurately identify optimal configurations based on their performance. This objective improves robustness to noise and runtime scale variance, which are common in early-stage accelerator performance data, as the model focuses on preserving relative orderings. This also enables us to efficiently integrate our cost model with a search technique to efficiently select the top-k (k-best) program configurations from the configuration space. Furthermore, prior work (Kaufman et al., 2021) has shown that training with ranking loss significantly improves a model's ability to identify optimal configurations. We use the *pairwise ranking loss* as our learning objective (implemented using margin ranking loss) to rank program configurations based on their true performance differences. For a given input matrix $M$, the *pairwise ranking loss* $(\mathcal{L})$ across all program configuration pairs can be defined as $\mathcal{L} = \sum_{(c_1,c_2)} \max(0, 1 - (\hat{r}_{M,c_1} - \hat{r}_{M,c_2})) \cdot \delta_{\text{true}}$; $\delta_{\text{true}} = \text{sign}(t_{M,c_1} - t_{M,c_2})$ where $\hat{r}_{M,c_1}$ and $\hat{r}_{M,c_2}$ are the predicted scores for configurations $c_1$ and $c_2$, respectively; $t_{M,c_1}$ and $t_{M,c_2}$ represent their actual runtimes; and $\delta_{\text{true}}$ signifies the true performance difference where $\text{sign}(x)$ returns 1 if $x > 0$, -1 if $x < 0$, and 0 if $x = 0$. This ensures that the model is penalized when the predicted ranking does not align with the true ranking. By minimizing this loss $(\mathcal{L})$, COGNATE improves its ability to accurately rank and identify the top-k program configurations. This also contributes to achieving our *accuracy objective* (Section A.2).

## B. Code Optimizations Across Hardware Platforms

- Loop strip-mining: Breaks down large software loops into smaller segments to optimize cache utilization.

- Loop reordering: Adjusts the execution order of loops to improve cache efficiency. Typically, it is applied after loop strip-mining.

- Format reordering: Reorganizes the data structure layout of sparsity patterns in memory to optimize memory access patterns

- Parallelization: Distributes tensor computations across multiple threads or processors to run tasks simultaneously.

- Loop binding: Assigns specific loop iterations to threads for parallel processing.

- Loop unrolling: Executes multiple iterations of a loop in a single iteration, reducing loop control overhead and boosting execution speed.

- Tiling: Decomposes a matrix into smaller blocks to optimize data reuse in the local memory and improve cache efficiency.

- Barrier: Applying a barrier would ensure all threads finish processing their current tile (synchronized) before progressing to the next stage.

- Cache bypassing: Capability of bypassing caches to to reduce cache pollution.

- Matrix reordering: Enhances data locality by reordering the input matrix.

## C. Generalizability

Our approximated code mappings are designed to enable well-established code optimizations that remain effective across emerging hardware platforms. For instance, we expect loop transformations (e.g. loop strip-mining, loop reordering, tiling, etc) to be implemented regardless of the underlying hardware platform although the exact implementation may be slightly different. Any hardware-specific optimizations (code optimizations that cannot be mapped) are included in the heterogeneous component using the latent encoder. Since the dimension of the latent embedding is fixed, we can effectively finetune using an already pre-trained model. To elaborate, let us have a qualitative discussion and explore the intuition behind incorporating approximate mapping of these loop transformations into sparse accelerators, using Intel PIUMA (Gerogiannis et al., 2024; Aananthakrishnan et al., 2023) and Vesper (Jin et al., 2024) as examples. These mappings are conceptually aligned with those we applied to SPADE and GPU, highlighting the general applicability of our approach across diverse hardware backends.

Intel PIUMA (Gerogiannis et al., 2024; Aananthakrishnan et al., 2023) is a configurable accelerator that has a RISC ISA making it CPU-programmable. This enables it to employ code optimizations that are available in CPUs. Hence, it is possible to implement SpMM and SDDMM sparse operations (kernels) with code optimizations such as loop reordering with a one-to-one mapping. Similarly, loop strip-ming and tiling can be mapped. However, similar to SPADE, where we accounted for the "barrier" optimization in the mapping process, one would need to consider the PIUMA "scratchpad reuse".

Vesper (Jin et al., 2024) is another reconfigurable accelerator designed for sparse computations, supporting three dataflow models implemented through distinct loop traversal orders. While the authors refer to the use of "tiling," they do not provide implementation details, source code, or a description of the tile size selection mechanism. Based on standard tiling practices, we can approximate Vesper's approach using loop stripping and loop reordering within our representation.

Intel PIUMA is proprietary, and Vesper's source code was not available. This made it infeasible to test our hypothesis on these accelerators. Hence, our current evaluation focuses only on two examples (SPADE and NVIDIA A100) primarily due to practical constraints. However, it should be emphasized that COGNATE was designed with hardware-agnostic principles in mind. We believe that as a wider range of accelerators becomes accessible to the research community, and as sparse compilation frameworks like SparseTIR (Ye et al., 2023) evolve, COGNATE can be extended with minimal changes.

Assuming these accelerators were available, the data collection process would still be highly time-consuming, likely requiring millions of machine hours to gather sufficient data for training, validation, and testing. For example, collecting performance data or training, validation, and testing across all matrices and experimental settings for SPADE required approximately 4 million CPU hours. Despite parallelizing experiments across multiple machines, each with 64 CPU cores, this process spanned nearly three months. While extending the evaluation to additional hardware platforms remains an important direction, it was beyond the practical scope of this work given resource constraints.

Further, frequent changes in emerging hardware may require updates to configuration mappings and fine-tuning, this challenge is significantly mitigated by our approach in COGNATE. As long as the entire kernel does not change or the newly introduced optimizations are heterogeneous, updating the mappings is relatively straight forward. However, relying solely on simulations would require rerunning them for a large number of configurations each time a change is made, resulting in significant computational and time costs. In contrast, our transfer learning-based approach significantly reduces the cost and time of running simulations. By collecting only a few data samples and fine-tuning the model, we can efficiently adapt to hardware changes without the need for extensive simulations. Hence, this approach not only reduces maintenance complexity but also accelerates the design process, making it more feasible to handle frequent and timely updates in emerging hardware.

## D. Cost Models for Early-Stage Sparse Accelerator Design

With the flexibility of recent accelerators to have software programmable kernels (Gerogiannis et al., 2023; Jin et al., 2024; Gerogiannis et al., 2024), integration of cost models and heuristics into the DSE pipeline has become an up and coming area (Gerogiannis et al., 2024; Jin et al., 2024). For example, Vesper is a recent work that had integrated an analytical model to a configurable sparse accelerator to enable higher throughput (Jin et al., 2024). HotTiles is another work that uses an analytical model to predict the performance of different accelerator processing elements (PEs) that accommodate intra-matrix heterogeneity (Gerogiannis et al., 2024). Further, in HotTiles, the authors acknowledged that a more accurate model could have enabled making better design decisions during the early stages. We believe that our proposed data-driven cost model framework, COGNATE, addresses this gap (resulting in speedups close to optimal) while complementing expert-driven strategies to enable more informed and better design decisions. This would effectively replace the analytical approaches with a data driven approach. The primary overhead associated with our approach arises from the need to gather data points to fine-tune the cost model. This overhead is minimal compared to the effort required for an expert to iteratively optimize a kernel for sparse workloads, where kernel performance is highly input-sensitive due to diverse sparsity patterns.

## E. An Example of Code Optimization Notations Used in Approximate Mappings

Here, we provide an additional explanation and an illustrative example to clarify the notation used in the code optimization mapping functions presented in Section 3.2. These are designed to approximate how high-level schedule configurations in SPADE are translated into low-level loop representations in CPU. The following example demonstrates how a sparse matrix-matrix multiplication (SpMM) configuration in SPADE is mapped into its corresponding loop-level representation using the defined notation. Consider the following high-level configuration for the SpMM operation in SPADE: `name, row_panels, column_panels, split, barrier, bypass, reorder, time = 144, 4, 1024, 1, 0, 0, 0, 38.83592`. Here, `row_panels`, `column_panels`, and `split` define the tiling strategy, while the binary flags `barrier`, `bypass`, and `reorder` indicate the use of additional code optimizations. Using our mapping functions, this configuration is mapped into the following loop-level representation: `name, i_split, j_split, k_split, loop_1, ..., loop_7, barrier, bypass, reorder, time = 144, 4, 1024, 32, 6, 7, 2, 4, 1, 3, 5, 0, 0, 0, 38.83592`. In this mapped form, the tiling parameters are converted to `i_split`, `j_split`, and `k_split`, which define how the loop indices are partitioned across the three dimensions. The sequence `loop_1` through `loop_7` encodes the execution order of the nested loops, and the binary flags are retained to preserve platform-specific scheduling decisions. This example demonstrates how our framework captures key aspects of tiling structure, loop ordering, and other scheduling optimizations.

## F. Hyperparamters

Table 3: Hyperparameters for model training/fine-tuning

| Hyperparameter | Value |
|---|---|
| Learning Rate | 0.0001 |
| Batch Size | 32 |
| Optimizer | Adam |
| Number of Epochs | 100 |
| Loss Function | MarginRankingLoss |

Table 4: Hyperparameters for the autoencoders

| Hyperparameter | Value |
|---|---|
| Learning Rate | 0.001 |
| Batch Size | 32 |
| Optimizer | Adam |
| Number of Epochs | 1000 |
| Loss Function | MSE |

Table 5: Composition of layers in the Input Featurizer ($\mathcal{IFE}$)

| Layer | Description |
|---|---|
| Layer 1 | MinkowskiConvolution (in_channels, 32, kernel_size=5) |
| Layer 2 | MinkowskiConvolution (32, 32, kernel_size=3) |
| Layer 3 | MinkowskiConvolution (32, 64, kernel_size=3) MinkowskiMaxPooling |
| Layer 4 | MinkowskiConvolution (64, 64, kernel_size=3) |
| Layer 5 | MinkowskiConvolution (64, 64, kernel_size=3) |
| Layer 6 | MinkowskiConvolution (64, 128, kernel_size=3) MinkowskiMaxPooling |
| Layer 7 | MinkowskiConvolution (128, 128, kernel_size=3) |
| Layer 8 | MinkowskiConvolution (128, 128, kernel_size=3) |
| Layer 9 | MinkowskiConvolution (128, 256, kernel_size=3)MinkowskiMaxPooling |
| Layer 10 | MinkowskiConvolution (256, 256, kernel_size=3) |
| Layer 11 | MinkowskiConvolution (256, 256, kernel_size=3) |
| Layer 12 | MinkowskiConvolution (256, 256, kernel_size=3) |
| Global Pooling Layer | MinkowskiGlobalAvgPooling |

Table 6: Composition of layers in the Predictor ($\mathcal{P}$)

| Component/Layer | Input Size | Output Size |
|---|---|---|
| Matrix Embedding (x) | 128 | 128 |
| Configuration Embedding (y) | 53 | 64 |
| Latent Embedding (z) | 64 | 64 |
| Concatenation (xyz) | 128 + 64 | 192 |
| Predictor Layer 1 | 192 | 128 |
| Predictor Layer 2 | 128 | 64 |
| Predictor Layer 3 | 64 | 1 |

# G. Additional Results

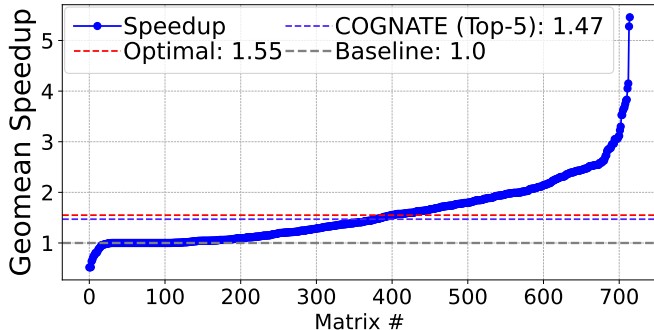

Figure 13: COGNATE (Top-5) per-matrix speedups (SpMM)

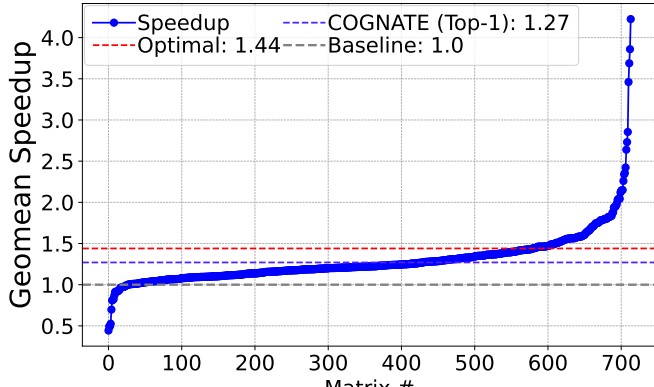

Figure 14: COGNATE (Top-1) per-matrix speedups (SDDMM)

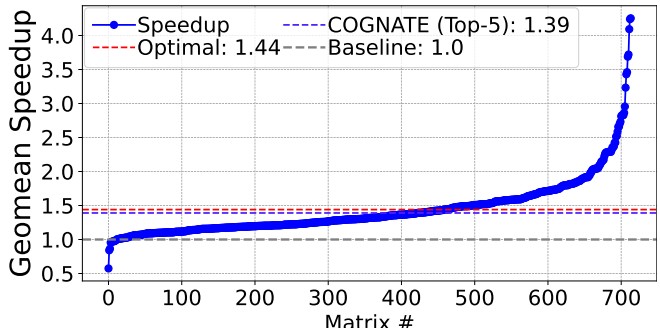

Figure 15: COGNATE (Top-5) per-matrix speedups (SDDMM)

