# OpenReview forum: "COGNATE: Acceleration of Sparse Tensor Programs on Emerging Hardware using Transfer Learning"
_ICML.cc/2025/Conference — ICML 2025 poster_

### Official Review · Reviewer_G9Vs · 2025-03-08

**Overall Recommendation:** 4

**Summary:**

The paper proposes a framework that trains a cost model for performance prediction on general-purpose hardware and then performs few-shot fine-tuning on emerging hardware accelerators. It focuses on optimizing sparse tensor programs on hardware accelerators. The proposed method achieves better hardware performance compared to other approaches like zero-shot and no-transfer learning. For instance, the source model is trained with data from 100 matrices, while the few-shot fine-tuning uses data from only 5 matrices. The paper claims that its few-shot learning approach can fine-tune the cost model with near-optimal accuracy using significantly smaller samples. These claims are generally supported by experimental data.

**Claims And Evidence:**

The paper asserts that its few-shot fine-tuning approach leads to near-optimal accuracy with significantly fewer samples. These claims are supported by empirical results, including performance comparisons against zero-shot and no-transfer approaches. The evaluation is rigorous and shows clear improvements in hardware performance across various platforms.

**Essential References Not Discussed:**

No.

**Experimental Designs Or Analyses:**

The experimental design includes adequate baselines and comparisons against different approaches based on the geometric mean speedup. The study also features good ablation experiments to show the impact of excluding individual components.

**Methods And Evaluation Criteria:**

The evaluation focuses on SpMM and SDDMM computations targeting three platforms: CPU, GPU, and an accelerator for sparse matrices. The baseline is adequate, and the dataset consists of 715 real-world matrices, which is sufficiently large. The proposed framework is compared with WacoNet, making the comparison fair.

**Other Comments Or Suggestions:**

* Clarifying the notation in Section 3.2 would greatly improve readability.
* Increasing the size of Figures 5 and 6 would help readers better understand the visualized data.

**Other Strengths And Weaknesses:**

Strengths:
* The paper presents an effective approach that improves upon WacoNet, which was the SOTA.
* It introduces a latent encoder to capture heterogeneous components and enhances the configuration mapper to optimize techniques unique to individual platforms, improving prediction accuracy.
* The empirical results demonstrate significant improvements in hardware performance.

Weaknesses:
* The clarity of some sections, particularly the equations in the code optimization section (Section 3.2), is lacking. The notations are not clearly explained and are difficult to follow.
* Some figures (e.g., Figures 5 and 6) are too small to read, which hinders the comprehension of the results.

**Questions For Authors:**

When the paper states, "The optimal speedup was determined by running all possible configurations (Section 4.2)," how many configurations are there?

**Relation To Broader Scientific Literature:**

The paper adequately cites state-of-the-art literature, including WacoNet, which was the previous SOTA method.

**Theoretical Claims:**

The paper does not make any theoretical claims.

---

> ### Author Rebuttal · Authors · 2025-04-01
>
> We sincerely thank the reviewer for recognizing the significance of the problem we address and the contributions of our work. We are especially grateful for the time and effort you dedicated to providing such a detailed and thoughtful review. We hope that our following response addresses your suggestions and questions.
>
> > ***Questions 01*** \
> > When the paper states, "The optimal speedup was determined by running all possible configurations (Section 4.2)," how many configurations are there?
>
> Thank you for pointing this out. We apologize for this incomplete statement. We will correct this to be read as:
>
> > “The optimal speedup was determined by running all possible configurations **within our defined search space**.”
>
> and, we will also provide more explanation of how the search space was constructed.
>
> To clarify, the search space we considered for the SPADE accelerator consists of a total of 256 configurations, derived from combinations of key tunable parameters related to tiling, barrier, cache bypassing, and matrix reordering.
>
> As shown in Table 1, the parameters for barrier, cache bypassing, and matrix reordering are binary (enabled or disabled). For tiling, we had three numerical parameters: row panels, column panels, and split factor. We had to select a constrained set of values for those numerical parameters since testing every single value would explode the size of the search space and would make data collection infeasible, as explained later.  We spaced those values to resemble the ones tested in the SPADE paper, as those are expected to show more significant performance deviations for different sparse matrices.
>
> In summary, our defined search space included 4 values for row panels [4, 32, 256, 2048], 4 values for column panels [1024, 16384, 65536, NUM_MATRIX_COLS] (where NUM_MATRIX_COLS depends on each matrix), and 2 values for the split factor [32, 256]. Combined with the three binary parameters, this resulted in 4 × 4 × 2 × 2 × 2 × 2 = 256 configurations.
>
> We opted for a constrained search space to make data collection feasible. Even so, collecting performance data for training, validation, and testing required approximately 4 million CPU hours. Although we parallelized the required experiments across 32 machines, each with 64 CPU cores, data collection for the constrained set still took nearly three months. Hence, data collection for the full theoretical search space would be infeasible. This challenge again reinforces the importance of data-efficient solutions like COGNATE.
>
> Once again, we thank the reviewer for raising this point, and we will include this clarification in the final version of the paper to provide additional context around our experimental setup.
>
> > ***Weakness 01*** \
> > The clarity of some sections, particularly the equations in the code optimization section (Section 3.2), is lacking. Clarifying the notation in Section 3.2 would greatly improve readability.
>
> We thank the reviewer for highlighting this valuable point. We will add more explanations in the final version. Further, we will include illustrative examples in the appendix to demonstrate how the equations are applied in practice. We believe these additions will make the technical content more accessible and further strengthen the presentation of our contributions.
>
> For instance, the following data point demonstrates how a SpMM schedule configuration in SPADE can be approximately mapped into its corresponding low-level loop configurations. The original high-level configuration specifies the key tunable parameters used in SPADE:
>
> > name, row_panels, column_panels, split,barrier, bypass, reorder, time \
> > 144, 4, 1024, 1, 0, 0, 0, 38.83592
>
> These values are mapped into the corresponding loop-level configuration by applying the equations defined in our paper. Specifically, row_panels, column_panels, and split are used to derive i_split, j_split, and k_split, which represent how the loop indices are partitioned across dimensions. The loop nest structure is encoded by loop_1 through loop_7, which define the execution order of the tiled loops. The binary flags barrier, bypass, and reorder are retained to reflect platform-specific scheduling configurations. The resulting mapped configuration looks like this:
>
> > name, i_split, j_split, k_split, loop_1, loop_2, loop_3, loop_4, loop_5, loop_6, loop_7, barrier, bypass, reorder, time \
> > 144, 4, 1024, 32, 6, 7, 2, 4, 1, 3, 5, 0, 0, 0, 38.83592
>
> This representation captures tiling structure, loop ordering, and other applicable optimizations. We will include similar examples along with the corresponding equations and code segments in the final version of the appendix.
>
> > ***Weakness 02*** \
> >  Some figures (e.g., Figures 5 and 6) are too small to read, which hinders the comprehension of the results.
>
> Thank you for pointing this out. We will update these figures to ensure all elements are legible and better support the interpretation of the results in the final version of the paper.

---

### Official Review · Reviewer_HuD5 · 2025-03-12

**Overall Recommendation:** 4

**Summary:**

This paper introduces COGNATE, a framework designed to optimize sparse tensor programs (e.g., SpMM and SDDMM) for emerging hardware accelerators using transfer learning. The key innovation lies in leveraging inexpensive data from general-purpose hardware (e.g., CPUs) to pre-train cost models and then fine-tuning them with minimal data from target accelerators. COGNATE addresses challenges such as heterogeneous program configuration spaces and high sample efficiency requirements by:

1. Approximate mapping of comparable code optimizations: Identifying homogeneous features across hardware platforms.
2. Latent encoding of hardware-specific features: Using autoencoders to compress heterogeneous configurations into a unified latent space.
3. Few-shot fine-tuning: Achieving competitive performance with only 5% of the data required by baseline methods.

Experimental results demonstrate significant speedups: $1.47\times$ (SpMM) and $1.39\times$ (SDDMM) on the SPADE accelerator, and $1.17\times$ (SpMM) and $1.15\times$ (SDDMM) on an NVIDIA A100 GPU, compared to existing techniques.

**Claims And Evidence:**

Yes.

**Essential References Not Discussed:**

As far as I know, there aren't Essential References Not Discussed.

**Experimental Designs Or Analyses:**

Since there is currently no cost model that leverages transfer learning to predict the performance of sparse tensor programs across different types of machines, this paper primarily compares the program performance obtained by various methods against the baseline.

From an experimental perspective, a comprehensive set of evaluations has been conducted, mainly including:

- Performance achieved by different transfer learning methods (Figure 4)
- Performance improvements contributed by individual components (Figure 7)
- Impact of different component choices on performance (Figure 9)
- Experiments on the data cost associated with transfer learning (Figures 10 and 12)

**Methods And Evaluation Criteria:**

The proposed methods and evaluation criteria, including the use of benchmark datasets, are generally sound and appropriate for the problem at hand. I did not identify any significant issues to address.

**Other Comments Or Suggestions:**

Since feature mapping requires manually identifying similar types of scheduling parameters and defining the mapping function, each time a new hardware platform emerges, a new mapping function needs to be designed. In contrast, the autoencoder-based latent representation proposed in this paper can automatically extract key information from the features. Have you considered applying the autoencoder to transform all features, rather than only the heterogeneous ones? If so, what were the results of this approach?

**Other Strengths And Weaknesses:**

Weaknesses:

- Strengths:
  - Clarity: Well-structured  with clear methodology.
- Weaknesses:
  - The types of sparse operations supported in this paper are somewhat limited, covering only SpMM and SDDMM. So the applicability to other sparse operations (e.g., convolution) is unclear. In contrast, WACO, which this paper compares against, also includes SpMV and MTTKRP.

**Questions For Authors:**

1. Have you conducted experiments on a broader range of sparse operations, such as SpMV and MTTKRP?
2. On GPUs, there are explicit data movement scheduling parameters, such as cache_read and cache_write. How did you handle these scheduling parameters in your approach? Did you treat them as homogeneous features that can be mapped across different hardware, or were they categorized as heterogeneous features and learned through the autoencoder?

**Relation To Broader Scientific Literature:**

The work builds on:

- ML-based cost models (WACO, TIR) for sparse tensor optimizations.
- Transfer learning techniques (e.g., feature augmentation/mapping) but classifies machine-specific features into homogeneous and heterogeneous categories for heterogeneous hardware.
- Hardware-aware optimizations (SPADE, HotTiles) by integrating learned models into accelerator design.

 The novelty lies in bridging sparse tensor program optimization and heterogeneous transfer learning, addressing the gap in early-stage accelerator development.

**Theoretical Claims:**

In Section 3.2, this paper employs a mapping function to illustrate the correlation between certain scheduling parameters across different types of machines, suggesting that they can be transformed into one another. However, in the discussion of loop strip mining, the explanation regarding the barrier on SPADE and the reorder operation on the CPU lacks clarity.

From my understanding, reordering on the CPU can involve multiple parameter combinations, whereas the barrier is a boolean parameter, making direct conversion between them infeasible.

Besides, there is an indexing error in the argument for loop reordering in the next paragraph.

---

> ### Author Rebuttal · Authors · 2025-04-01
>
> We sincerely thank the reviewer for recognizing the significance of the problem we address and the contributions of our work. We are especially grateful for the time and effort you dedicated to providing such a detailed and thoughtful review. We hope that our following response addresses your suggestions and questions.
>
> > ***Weakness 01 & Question 01*** \
> > The types of sparse operations supported in this paper are somewhat limited, covering only SpMM and SDDMM. … Have you conducted experiments on a broader range of sparse operations, such as SpMV and MTTKRP?
>
> We thank the reviewer for this important question. We agree that our current scope considers only SpMM and SDDMM sparse operations. This limitation stems from the fact that these are the only sparse operations currently natively supported by the SPADE accelerator [1] and the SparseTIR framework [2].
>
> That said, it is possible to indirectly express SpMV in SPADE using a workaround if the operation is expressed as a SpMM with a very skinny zero-padded dense matrix. During our data collection process, we had gathered performance numbers for a split factor of 16 (an SpMM with 256 dense columns is broken down into 16 SpMMs, each with 16 dense columns). Each of these smaller SpMMs is computationally equivalent to a zero-padded SPADE SpMV. Since we had these performance numbers already available, we trained a model for SpMV during the rebuttal period to evaluate COGNATE’s ability to generalize. The results were promising, with COGNATE achieving a 1.25× geometric mean speedup (top-1) prediction, compared to the optimal speedup of 1.36×. These findings demonstrate that COGNATE could potentially generalize to SpMV-style workloads. We will include these findings and the above clarification in the final version of the paper.
>
> > ***Other Comments*** \
> > ... Have you considered applying the autoencoder to transform all features, rather than only the heterogeneous ones? If so, what were the results of this approach?
>
> We thank the reviewer for this insightful question. We share your concern about the need to have mapping functions as diverse new hardware platforms emerge. However, our results suggest that the inclusion of these enabled us to overcome the limitations of prior work in the domain, which achieved effective knowledge transfer only between hardware platforms of the same architecture.
>
> We explored the idea of applying the autoencoder to all features (both homogeneous and heterogeneous), but found that this approach performed poorly in our experiments compared to the solution we propose in COGNATE. Specifically, we observed the following results for SpMM for the SPADE accelerator when using the autoencoder to encode all input features;
>
> - Top-1 speedup: 1.118× (comapred to 1.40× in COGNATE)
> - Top-5 speedup : 1.237× (comapred to 1.47× in COGNATE)
>
> Our results suggest that transforming homogeneous features, which tend to be consistent and interpretable across hardware platforms, via an autoencoder along with the rest of the features could introduce unnecessary complexity, which hinders the model's ability to generalize effectively. By contrast, limiting the autoencoder to heterogeneous features allows us to preserve the generalizability of shared input characteristics. We will include these findings in the final version to clarify our design choices.
>
> > ***Question 02*** \
> > On GPUs, there are explicit data movement scheduling parameters, such as cache_read and cache_write. How did you handle these scheduling parameters in your approach? ...
>
> We thank the reviewer for pointing this out. In our current experiments, we followed the default behavior used in SparseTIR examples, where cache_read and cache_write scheduling optimizations are enabled for SDDMM and disabled for SpMM. As a result, we did not explicitly include these parameters as part of the search space in our evaluation. However, if we were to expand the search space to include these scheduling parameters, we agree that they would need to be treated as heterogeneous features, since they represent architecture-specific optimizations primarily exposed in GPU environments. In that case, they would be encoded through the autoencoder, rather than mapped as homogeneous features. We will explore the possibility of expanding the search space to incorporate these optimizations into our experiments in the final version of the paper.
>
> [1] Gerogiannis, Gerasimos, et al. "Spade: A flexible and scalable accelerator for spmm and sddmm." Proceedings of the 50th Annual International Symposium on Computer Architecture. 2023.
>
> [2] Ye, Zihao, et al. "Sparsetir: Composable abstractions for sparse compilation in deep learning." Proceedings of the 28th ACM International Conference on Architectural Support for Programming Languages and Operating Systems, Volume 3. 2023.

---

> > ### Comment · Reviewer_HuD5 · 2025-04-02
> >
> > Thanks for the efforts and clarification. Overall, I still like this paper and will keep my score.

---

> > > ### Author Response · Authors · 2025-04-09
> > >
> > > We sincerely thank the reviewer for acknowledging our efforts and clarifications. We appreciate the reviewer’s positive assessment of our work.

---

### Official Review · Reviewer_eqda · 2025-03-13

**Overall Recommendation:** 4

**Summary:**

The submission introduces COGNATE, a novel framework designed to optimize sparse tensor programs on emerging hardware accelerators using machine learning-based cost models. It addresses the challenges of optimizing sparse tensor programs, such as Sparse Matrix-Matrix Multiplication (SpMM) and Sampled Dense-Dense Matrix Multiplication (SDDMM), on early-stage accelerators where performance is sensitive to sparse input variations and data collection via simulators is costly. COGNATE leverages transfer learning by pre-training cost models on inexpensive CPU data and fine-tuning them with minimal accelerator-specific data (5% of typical requirements), achieving high sample efficiency.

Key contributions include: (1) techniques to segregate program configurations into homogeneous (mapped via approximate code optimization mappings) and heterogeneous (encoded via autoencoders) components, enabling effective knowledge transfer across hardware platforms; (2) a data-frugal cost model framework that modifies the WACO architecture for better transferability, incorporating a configuration mapper, enhanced input featurizer, latent encoder, and predictor; and (3) extensive evaluations demonstrating COGNATE’s effectiveness. Main results show COGNATE achieves average speedups of 1.47× (up to 5.46×) for SpMM and 1.39× (up to 4.22×) for SDDMM on the SPADE accelerator, and 1.17× (up to 1.61×) for SpMM and 1.15× (up to 1.49×) for SDDMM on an NVIDIA A100 GPU, outperforming existing transfer learning methods by 28.44% on SPADE. These findings highlight COGNATE’s ability to deliver near-optimal performance with minimal data overhead, enhancing design space exploration for emerging sparse accelerators.

**Claims And Evidence:**

The claims in the submission are generally supported by clear and convincing evidence, including detailed experimental results, figures, and tables comparing COGNATE’s performance against baselines and alternative methods. The main findings—speedups of 1.47× for SpMM and 1.39× for SDDMM on SPADE, and 1.17× and 1.15× on A100 GPU—are backed by evaluations on 715 real-world matrices from the SuiteSparse collection, with geomean speedups, per-matrix speedups (Figures 5, 13-15), and accuracy metrics (Figure 6) provided. The claim of data efficiency (using 5% of typical data) is substantiated by comparisons of data collection overhead (Figure 10) and fine-tuning sample analysis (Figure 12). Ablation studies (Figure 7) and component design choices (Figures 8-9) further support the algorithmic innovations.

However, two claims could be seen as less robustly evidenced:

- Generalizability to Other Accelerators: The paper claims COGNATE is generalizable beyond SPADE and A100, referencing Intel PIUMA and Vesper (Section C). However, no experimental data is provided for these platforms due to proprietary restrictions or unavailable source code, weakening this claim with speculative rather than empirical support.

- Near-Optimal Accuracy: The assertion of "near-optimal accuracy" (e.g., 95% of optimal speedup on SPADE) relies on comparing COGNATE’s top-5 predictions to an optimal baseline derived from exhaustive configuration testing. While results are strong, the lack of a statistical significance test or error bounds around the 95% figure undermines the claim.

**Essential References Not Discussed:**

- Sparse Accelerator Design Context:
Missing: "OuterSPACE" by Parashar et al. (MICRO 2017) introduced a sparse tensor accelerator with configurable tiling, relevant to COGNATE’s mapping of optimizations like tiling and barriers.

- Transfer Learning Efficiency:
Missing: "MetaTune" by Lee et al. (MLSys 2023) proposed a meta-learning approach for tuning tensor programs across platforms with minimal data, akin to COGNATE’s few-shot fine-tuning.

- Cost Model Accuracy:
Missing: "AutoTVM" by Chen et al. (OSDI 2018) (beyond the cited TVM paper) detailed a tuning framework with cost models for sparse workloads, achieving near-optimal schedules.

**Experimental Designs Or Analyses:**

Dataset, Training, and Evaluation Setup (4.1):
- Design: Uses SuiteSparse Matrix Collection (1500 matrices for training, 715 for evaluation) across CPU, SPADE, and A100 GPU, with 100 random configurations per matrix.
- Assessment: Sound dataset choice; widely accepted benchmark. Random sampling ensures diversity. Separation of training and evaluation sets avoids overlap. No major issues.

Transferability to SPADE (4.2):
- Design: Compares COGNATE (Top-1/Top-5) speedups vs. baselines (zero-shot, no-transfer, WACO+FA/FM) using geomean speedups.
- Assessment: Valid comparison with clear metrics. Optimal speedup baseline (exhaustive testing) is a strong reference. Fine-tuning with 5 matrices is justified later (4.4). No significant flaws.

Transferability to GPU (4.3):
- Design: Extends evaluation to A100 GPU, comparing COGNATE to cuSPARSE and modified WACO models.
- Assessment: Logical extension to test generalizability. Consistent methodology with SPADE. cuSPARSE as a baseline is relevant. No issues.

**Methods And Evaluation Criteria:**

Yes, the proposed methods and evaluation criteria in COGNATE are well-suited to the problem of optimizing sparse tensor programs on emerging hardware accelerators. The transfer learning approach—pre-training on CPU data and fine-tuning with minimal accelerator data—addresses the high cost of simulator-based data collection and input sensitivity of sparse operations like SpMM and SDDMM. Key components (configuration mapper, latent encoder, enhanced input featurizer) logically tackle the heterogeneity and data efficiency challenges outlined.

The evaluation criteria, including geomean speedups, top-1/top-5 predictions, and accuracy metrics (Pairwise Ranking Loss, Ordered Pair Accuracy, Kendall’s Tau), effectively measure performance and ranking quality against baselines (WACO+FA/FM, no-transfer models). The use of the SuiteSparse Matrix Collection (715 real-world matrices) as a benchmark dataset is appropriate, offering a diverse, established standard for sparse tensor research. Testing on SPADE and A100 GPU aligns with the focus on emerging and established accelerators. Overall, the methods and criteria are sensible and relevant to the application.

**Other Comments Or Suggestions:**

Could use a brief table summarizing key hyperparameters for autoencoders (like Table 3 for main model) to improve reproducibility.

**Other Strengths And Weaknesses:**

Strengths:
- Originality: Creatively combines transfer learning, feature segregation, and latent encoding to address sparse tensor optimization on emerging accelerators, a novel synthesis of ideas from Neyshabur et al. (2020) and Won et al. (2023).
- Significance: Tackles a real-world bottleneck in early-stage accelerator design, offering a practical, data-frugal solution with significant speedups (up to 5.46×), impactful for hardware-software co-design.
- Clarity: Well-structured with clear figures (e.g., Figure 3) and detailed appendices, making complex concepts accessible.

Weaknesses:
- Originality: While innovative, it heavily builds on WACO (Won et al., 2023), potentially limiting its perceived novelty in the ML optimization space.
- Significance: Generalizability to untested accelerators (e.g., PIUMA, Vesper) is speculative, tempering its broader impact claims.

**Questions For Authors:**

How was the 5-matrix fine-tuning sample size chosen beyond empirical observation (e.g., statistical power analysis), and how sensitive are results to slight variations (e.g., 3 or 7 matrices)?

**Relation To Broader Scientific Literature:**

- Segregation and Encoding of Program Configurations:
Builds on transfer learning concepts from Weiss et al. (2016) and Zhuang et al. (2020), extending homogeneous transfer (e.g., Sasaki et al., 2022) to heterogeneous CPU-to-accelerator settings.
Feature reuse and latent encoding draw from Neyshabur et al. (2020), adapting them to sparse tensor optimization, unlike prior feature augmentation (Daumé III, 2009; Duan et al., 2012) that struggled with sparsity.

- Data-Frugal Cost Model Framework:
Enhances WACO (Won et al., 2023) by refining its SCNN-based architecture (Graham & Van der Maaten, 2017) for transferability, contrasting with data-intensive models like Ansor (Zheng et al., 2020).
Aligns with few-shot learning ideas (Shen et al., 2021), reducing data needs compared to traditional ML-based optimization (Chen et al., 2018b; Baghdadi et al., 2021).

- Evaluation on Sparse Accelerators:
Extends sparse tensor optimization from CPU/GPU systems (Kjolstad et al., 2017; Ye et al., 2023) to emerging accelerators like SPADE (Gerogiannis et al., 2023), complementing hardware-specific efforts (Hegde et al., 2019; Li et al., 2023).
Speedup results (1.47× SpMM, 1.39× SDDMM) improve on prior sparse kernel optimizations (Hong et al., 2019; Jiang et al., 2020), offering a data-driven alternative to analytical models (Jin et al., 2024).

COGNATE bridges ML-based program optimization and hardware acceleration, advancing sample efficiency and cross-platform applicability beyond prior works.

**Theoretical Claims:**

The submission primarily focuses on empirical contributions rather than theoretical proofs, so there are no formal mathematical proofs to check for correctness.

---

> ### Author Rebuttal · Authors · 2025-04-01
>
> We sincerely thank the reviewer for recognizing the significance of our contributions. We are especially grateful for the time and effort you dedicated to providing such a detailed and thoughtful review. We also appreciate the additional references you shared, which we'll include in the final version. We hope that our following response addresses your suggestions and questions.
>
> > ***Other Comments*** \
> > ... key hyperparameters for autoencoders … to improve reproducibility.
>
> We thank the reviewer for pointing this out. The hyperparameters for the autoencoder training are:
>
> - Learning Rate: 0.001
> - Loss: MSE
> - Optimizer: Adam
> - Batch Size: 32
> - Epochs: 1000
>
> We will include this information in a table similar to Table 3.
>
> > ***Question 01*** \
> > ... how sensitive are results to slight variations (e.g., 3 or 7 matrices)?
>
> We thank the reviewer for this thoughtful question. The choice of 5 matrices for fine-tuning was primarily guided by empirical observation and the need to balance transfer learning effectiveness with the cost of data collection. As mentioned in the paper, “the matrices were randomly selected from the training set while ensuring a balanced representation of their dimensions and sparsity.” To do this, we first divided the available matrices into five groups based on input size [8192, 32768, 65536, 131072, and > 131072], then randomly sampled a matrix from each group while ensuring the selected subset captured a reasonable range of sparsity. We used the same randomly selected matrices for the non-transfer learning baselines to ensure consistency and fair comparison across experimental settings.
>
> In response to the reviewer’s suggestion, we conducted additional experiments for SpMM using 3 and 7 matrices to assess sensitivity to small variations in fine-tuning set size. The observed top-1 speedups were (compared to 1.40×);
>
> - 3 matrices: 1.301×
> - 7 matrices: 1.405×
>
> These results suggest that COGNATE is relatively robust to small variations in the size of the fine-tuning dataset. While using only 3 matrices results in a slight drop in performance, the model still achieves meaningful gains over the zero-shot baseline. Finally, we acknowledge that no formal statistical power analysis was used in selecting the fine-tuning size. We view this as a promising direction for future work.
>
> > ***Weakness 01*** \
> > While innovative, it heavily builds on WACO …, potentially limiting its perceived novelty in the ML optimization space.
>
> We thank the reviewer for this observation. We agree that our work builds on top of WACO, the current state of the art in ML-based sparse autotuning. However, the key contributions of COGNATE are distinct and extend beyond WACO in several important ways. While we adopt WACO’s search space and program representation due to their relevance to sparse tensor programs, our primary focus is cross-platform generalization and efficient model transfer, which are orthogonal to WACO’s contributions. Specifically, COGNATE’s framework leverages the homogeneity of input features across hardware platforms while mitigating heterogeneity to efficiently fine-tune learned cost models for accelerating sparse operations on emerging hardware. As part of future work, we are actively exploring the application of COGNATE to other domains beyond sparse tensor programs and WACO’s framework.
>
> > ***Weakness 02*** \
> > Generalizability to untested accelerators (e.g., PIUMA, Vesper) is speculative, ...
>
> We acknowledge the reviewer’s concern. Our current evaluation focuses on two examples (SPADE and NVIDIA A100) primarily due to practical constraints, including lack of access to closed-source platforms like PIUMA and Vesper. Even if we had access, the data collection process for these accelerators would be highly time-consuming, likely requiring millions of machine hours to collect sufficient data for training, validation, and testing. For instance, collecting performance data for training, validation, and testing for SPADE required approximately 4 million CPU hours.  Although we parallelized the required experiments across 32 machines, each with 64 CPU cores, this process took us nearly three months.  Hence, while extending our evaluation to even more hardware platforms would be desirable, it was not feasible given the computational resources and simulators we had access to.
>
> However, we emphasize that COGNATE was designed with hardware-agnostic principles in mind. We believe that as a wider range of accelerators becomes accessible to the research community, and as sparse compilation frameworks like SparseTIR evolve, COGNATE can be extended with minimal changes. We will further clarify this point in the final version of the paper. That being said, in the final version, we will tone down our generalization claims and more clearly state that the quantitative demonstration of COGNATE’s even broader applicability (beyond the two hardware accelerators we evaluated) remains an important direction for future work.

---

> > ### Comment · Reviewer_eqda · 2025-04-07
> >
> > Acknowledged

---

> > > ### Author Response · Authors · 2025-04-09
> > >
> > > We sincerely thank the reviewer for acknowledging our efforts and clarifications.

---

### Official Review · Reviewer_cfbi · 2025-03-15

**Overall Recommendation:** 2

**Summary:**

The paper introduces COGNATE, a novel framework for developing learned cost models to optimize sparse tensor programs on emerging hardware platforms. COGNATE leverages transfer learning to adapt cost models from general-purpose hardware (e.g., CPUs) to specialized accelerators with minimal fine-tuning data.

Main Contributions:
- Transfer Learning Approach: COGNATE uses a pre-trained model on CPU data and fine-tunes it on sparse accelerators, significantly reducing the need for expensive simulation data.
- Homogeneity and Heterogeneity Handling: It maps similar code optimizations across platforms and uses autoencoders to encode heterogeneous components, allowing for efficient knowledge transfer.
- Performance: COGNATE achieves average speedups of 1.47× for Sparse Matrix-Matrix Multiplication (SpMM) and 1.39× for Sampled Dense-Dense Matrix Multiplication (SDDMM) on the SPADE accelerator.

Update after rebuttal:

I would like to sincerely thank the authors for taking the time to respond to all the issues I raised. However, as I pointed out in the initial review, the design and evaluation are quite limited with SPADE and NVIDIA A100 GPU. The authors claimed that just focusing on two examples (SPADE and NVIDIA A100) is primarily due to practical constraints, such as lack of access to closed-source platforms. However, there are plenty of better platforms than closed-source platforms that can be accessed, just to name a few: NVIDIA Jetson AGX Orin, NVIDIA H20 & H100 GPU. Moreover, from the authors' replies, the effectiveness of the proposed COGNATE method is highly related to the hyper-parameter settings. For example, the size of the dataset will significantly influence the transfer effect. So, it brings the concern of the generalization of the COGNATE method. Finally, the Sparse Tensor Core features may change significantly for different accelerator hardware platforms. For example, H100 GPU further supports the new FP8 data type, and the combination of Tensor Memory Accelerator (TMA). The COGNATE method also fails to address how to fit the 2:4 structured sparse pattern, which is the most important feature introduced in the NVIDIA A100 sparse Tensor Core. With all these concerns, I still hold my initial score. So, the current version may not meet the acceptance threshold for ICML.

**Claims And Evidence:**

The claims made in the submission are generally supported by clear and convincing evidence, but there are a few aspects that could be scrutinized further:
1. Performance Comparisons:
- Evidence: The paper provides extensive comparisons with existing techniques like WACO+FA and WACO+FM, showing that COGNATE outperforms them by a significant margin (28.44% improvement for SpMM on SPADE)
- Potential Issue: While the comparisons are thorough, it would be beneficial to see more detailed analysis on why COGNATE performs better, especially in terms of its ability to handle heterogeneity and its data efficiency

2. Data Efficiency:
- Evidence: COGNATE is shown to achieve comparable performance with only 5% of the data required by accelerator-specific models, which is a significant reduction in data collection overhead
- Potential Issue: The paper could further elaborate on how this efficiency is achieved, particularly in terms of the autoencoder's role in compressing heterogeneous features and the impact of using a reduced number of layers in the cost model

3. Generalizability:
- Evidence: COGNATE demonstrates its applicability across different hardware platforms, including SPADE and NVIDIA A100 GPU, with notable speedups in both cases
- Potential Issue: While the results are promising, additional evaluations on more diverse hardware platforms would strengthen the claim of generalizability

**Essential References Not Discussed:**

No

**Experimental Designs Or Analyses:**

The experimental design and analyses in the paper on COGNATE appear to be sound, but there are a few aspects that could be scrutinized further:
- The evaluations on more diverse hardware platforms (e.g., other specialized accelerators, FPGAs) could strengthen the claim of generalizability.
- Further analysis on how the number of fine-tuning samples affects performance could provide deeper insights into the limits of COGNATE's data efficiency.
- Including other metrics (e.g., energy efficiency, memory usage) could offer a more holistic evaluation of COGNATE's benefits.

**Methods And Evaluation Criteria:**

The proposed methods and evaluation criteria in the paper make sense for the problem of optimizing sparse tensor programs on emerging hardware platforms. The paper addresses a critical challenge in optimizing sparse tensor programs, which are essential in deep learning and graph analytics. The use of real-world sparse matrices from the SuiteSparse Matrix Collection provides a comprehensive and realistic evaluation setup.

**Other Comments Or Suggestions:**

N/A

**Other Strengths And Weaknesses:**

The paper could provide more discussion on the broader impact of COGNATE beyond the specific hardware platforms evaluated (SPADE and NVIDIA A100 GPU). Demonstrating its applicability to a wider range of accelerators or scenarios could enhance its significance.

Some sections, particularly those detailing the mapping functions and autoencoder training, could benefit from additional illustrations or step-by-step explanations to improve clarity for readers unfamiliar with these techniques. Additionally, the paper assumes a strong background in sparse tensor programs and transfer learning, which might limit accessibility for readers from other domains.

The paper could provide more details on the hyperparameter tuning process and how specific hyperparameters were chosen. This would help in reproducing the results and understanding the sensitivity of COGNATE to different hyperparameter settings.

**Questions For Authors:**

1. How do you envision extending COGNATE to support a broader range of emerging hardware platforms beyond SPADE and NVIDIA A100 GPU? Are there specific challenges or modifications needed for other types of accelerators?

2. You mention that using a moderate-sized dataset for the source model helps mitigate negative transfer. Can you provide more insights into how the size of the source dataset affects the fine-tuning performance on target platforms?

3. As the complexity of sparse tensor programs increases, how does COGNATE's performance scale? Are there any plans to address more complex operations or larger-scale programs?

4. Given the rapid evolution of hardware, how does COGNATE adapt to changes in hardware architecture or new optimizations introduced in emerging accelerators?

5. Can you elaborate on the pairwise ranking loss used in training the cost model? How does this objective function contribute to the model's ability to identify optimal program configurations?

**Relation To Broader Scientific Literature:**

The key contributions of the paper on COGNATE are closely related to the broader scientific literature in several ways:
1. Transfer Learning for Cost Models:
Prior Work: Transfer learning has been successfully applied in various domains to reduce data requirements for target tasks (Weiss et al., 2016; Zhuang et al., 2020). In program optimization, transfer learning has been used to adapt cost models across similar hardware platforms (Sasaki et al., 2022; Zheng et al., 2021).

2. Handling Heterogeneity in Transfer Learning:
Prior Work: Existing heterogeneous transfer learning techniques, such as feature augmentation and feature mapping, have limitations when dealing with diverse program configurations across different hardware platforms (Daumé III, 2009; Duan et al., 2012).

3. Optimization of Sparse Tensor Programs:
Prior Work: Optimizing sparse tensor programs is crucial for deep learning and graph analytics, with techniques like TACO (Kjolstad et al., 2017) and WACO (Won et al., 2023) providing significant performance improvements.

4. Data Efficiency in Early-Stage Hardware Development:
Prior Work: The high cost of collecting large datasets for emerging hardware platforms is a significant challenge (Gerogiannis et al., 2023).

**Theoretical Claims:**

The paper on COGNATE does not present formal proofs for its theoretical claims. Instead, it focuses on empirical evaluations and algorithmic design to support its contributions. However, there are no explicit mathematical proofs provided in the paper for theoretical claims like:
- Optimality of the Transfer Learning Approach: The paper demonstrates empirically that COGNATE outperforms other transfer learning techniques but does not provide a theoretical proof of optimality.
- Effectiveness of Latent Encoding: The use of autoencoders to handle heterogeneous components is shown to be effective in practice, but there is no formal proof of its theoretical advantages over other methods like feature augmentation.
- Data Efficiency: While the paper shows that COGNATE achieves comparable performance with significantly less data, there is no formal proof that this approach is optimal in terms of data efficiency.

---

> ### Author Rebuttal · Authors · 2025-04-01
>
> We sincerely thank the reviewer for recognizing the significance of the problem we address and for acknowledging the contributions of our work. We are especially grateful for the time and effort you invested in providing a detailed and thoughtful summary of the paper’s strengths and areas for improvement. We hope that our following response addresses suggestions and questions.
>
> **Q1 (Question) & W1 (Weakness)**: We thank the reviewer for this thoughtful question. In Appendix Section C (Generalizability), we provide an initial qualitative discussion and intuition on how COGNATE can be generalized. Our current evaluation focuses on two examples (SPADE and NVIDIA A100) primarily due to practical constraints, such as lack of access to closed-source platforms (PIUMA and Vesper). Even if we had access, the data collection for these accelerators would be highly time-consuming, requiring millions of machine hours. For instance, collecting performance data for SPADE required approximately 4 million CPU hours. Hence, while extending our evaluation to more hardware platforms would be desirable, it was not feasible given the computational resources and simulators we had access to. We believe that as a wider range of accelerators becomes accessible to the research community, COGNATE can be extended with minimal changes. We will further clarify this point in the final version of the paper.
>
> **W2 & W3**: We thank the reviewer for pointing this out. In the final version, we will improve the content to enhance clarity and include more details about hyperparameter tuning.
>
> **Q2**: We thank the reviewer for this insightful question. We explored the effects of negative transfer by training the source model using data samples with 5, 20, 500, and 1000 matrices (Figure 11). While a larger dataset improves generalization in the source domain, it may also cause over-specialization to the source platform. This overfitting hampers fine-tuning, as specialized features may not transfer well to the target platform. Our empirical findings show that training the source model on 100 matrices strikes a good balance, capturing useful patterns while maintaining generality for transfer. This reduces the effort needed during fine-tuning, leading to faster convergence and better performance. This is especially valuable given the high cost of collecting data for SPADE.
>
> **Q3**: We thank the reviewer for this important question. Our current evaluation focuses on SpMM and SDDMM because these are the sparse operations currently natively supported by the SPADE and SparseTIR. That said, SpMM and SDDMM serve as foundational blocks for complex programs. To evaluate COGNATE’s scalability, we conducted preliminary experiments during the rebuttal period on an end-to-end GNN workload running on a GPU, using the ‘transient’ sparse matrix from our testset as input (178,866 nodes with 961,368 non-zeros) and GraphSAGE model. The model was configured with 3 hidden layers, 256 hidden features. COGNATE achieved notable speedups over the default SparseTIR implementation, with a 1.30× speedup for inference and 1.28× for training, demonstrating the scalability of COGNATE. We will include this result in the final version of the paper and highlight it as a key direction moving forward.
>
> **Q4**: We appreciate the reviewer raising this point. While changes in emerging accelerators may necessitate updates to configuration mappings or model parameters, COGNATE significantly reduces this burden by relying on lightweight fine-tuning, rather than requiring retraining from scratch. As long as the overall structure of the sparse tensor program remains consistent, COGNATE can quickly adapt by using a small number of new performance samples. In contrast, traditional cost model construction approaches would require re-evaluating a large number of configurations. We provided further elaboration on this in under Generalizability (Section C) in the Appendix.
>
> **Q5**: We appreciate the reviewer’s interest in the training objective. Our goal during model training is not to predict the absolute runtime of a sparse tensor program configuration, but rather to rank candidate configurations by their relative performance, enabling the selection of the best-performing ones. To this end, we adopted a pairwise ranking loss, which is more aligned with the optimization task than pointwise losses. This objective improves robustness to noise and runtime scale variance, which are common in early-stage accelerator performance data, as the model focuses on preserving relative orderings. Furthermore, prior work (Kaufman et al., 2021) has shown that training with ranking loss significantly improves a model’s ability to identify optimal configurations. We will clarify this motivation and include these details in the final version to make the learning objective and its impact more explicit. Further, if space permits, we will move the loss equation (in Appendix Section A.4) to the main text of the paper.

---

### Decision · Program_Chairs · 2025-05-01

**Decision:**

Accept (poster)

**Comment:**

The authors generally found the methodology clear, with clear novelty differentiating it to comparable the closets existing approaches in the literature, and appropriate baselines of comparison. Significant speedups are demonstrated on both the SPADE and A100 GPU accelerators, the method is data efficient, and the paper relatively well written.

While three of the reviewers are in favour of acceptance of the work post-rebuttal, one reviewer advocates rejection. The primary concern of the reviewers on the negative side is based in the main motivation of the proposed methodology: the method's ability to generalize to different/emerging hardware accelerators. Specifically, two of the reviewers' concerns centred around the evaluation on only two hardware platforms, which does appear to be a significant limitation on the author's claim of generalizability. While the authors address this in the rebuttal as a lack of availability of other hardware architectures, reviewers did not find this convincing, pointing out there are other platforms available to researchers or even simply different GPU architectures that would allow the authors to better demonstrate this strong claim holds.